# Revisiting Logit Distributions for Reliable Out-of-Distribution Detection

**Jiachen Liang**[1,2]    **Ruibing Hou**[1*]    **Minyang Hu**[1,2]    **Hong Chang**[1,2]    **Shiguang Shan**[1,2]

**Xilin Chen**[1,2]

[1] State Key Laboratory of AI Safety, Institute of Computing Technology, CAS, China
[2] University of Chinese Academy of Sciences (CAS), China
{jiachen.liang, minyang.hu}@vipl.ict.ac.cn, {houruibing, changhong, sgshan, xlchen}@ict.ac.cn

## Abstract

Out-of-distribution (OOD) detection is critical for ensuring the reliability of deep learning models in open-world applications. While post-hoc methods are favored for their efficiency and ease of deployment, existing approaches often underexploit the rich information embedded in the model's logits space. In this paper, we propose LogitGap, a novel post-hoc OOD detection method that explicitly exploits the relationship between the maximum logit and the remaining logits to enhance the separability between in-distribution (ID) and OOD samples. To further improve its effectiveness, we refine LogitGap by focusing on a more compact and informative subset of the logit space. Specifically, we introduce a training-free strategy that automatically identifies the most informative logits for scoring. We provide both theoretical analysis and empirical evidence to validate the effectiveness of our approach. Extensive experiments on both vision-language and vision-only models demonstrate that LogitGap consistently achieves state-of-the-art performance across diverse OOD detection scenarios and benchmarks. Code is available at https://github.com/GIT-LJc/LogitGap.

## 1 Introduction

Deep learning models have demonstrated remarkable success across various computer vision tasks, typically under the closed-set assumption. However, this assumption often fails to hold in real-world applications such as autonomous driving, medical imaging, and access control systems. In open-world scenarios, deployed models are inevitably exposed to out-of-distribution (OOD) samples, which can result in unreliable predictions, thereby introducing significant risks to the safety and reliability of the system. To mitigate these risks, OOD detection has been proposed to detect and reject such OOD inputs before making decisions.

In recent years, the post-hoc OOD detection methods have attracted significant attention. These methods directly operate on pre-trained models without modifying their parameters [30, 14, 59, 25, 29, 45, 17], offering high deployment flexibility and computational efficiency. A core problem for the post-hoc methods is how to design a scoring function that effectively maximize the separability between ID and OOD samples. Two representative scoring functions are Maximum Logit (MaxLogit) [14] and Maximum Concept Matching (MCM[2][33]). MaxLogit simply uses the largest logit value as the OOD score, which completely disregards information from the remaining logits. In contrast,

---

[*]Corresponding author

[2]Although MCM [33] and Max Softmax Probablity (MSP [17]) target different objectives, their methods for computing OOD scores are equivalent. Hence, we analyze only one in detail.

39th Conference on Neural Information Processing Systems (NeurIPS 2025).

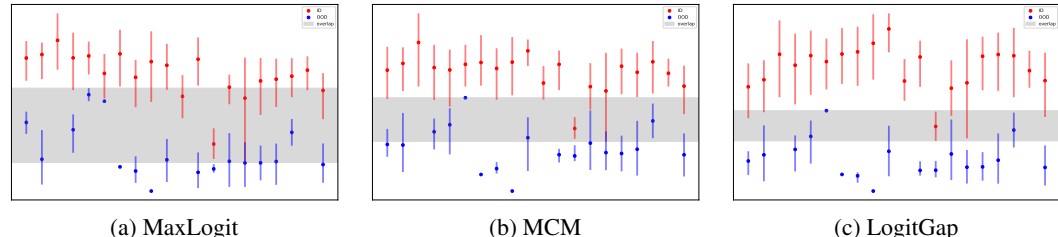

| (a) MaxLogit | (b) MCM | (c) LogitGap |

Figure 1: OOD score distributions for ID and OOD samples across scoring functions. The x-axis shows predicted class indices. Dots represent mean scores per class; vertical lines indicate score ranges. Gray shading marks overlap regions of ID and OOD scores. Results are reported using CLIP ViT-B/16 model on ImageNet-20 (ID) and ImageNet-10 (OOD).

MCM applies a softmax function into the logits and then takes the maximum softmax probability as the OOD score. By applying a softmax function, MCM significantly reducing the overlap between the score distributions of ID and OOD samples (Figure 1). However, the softmax function used in MCM ignores the absolute magnitudes of original logits, resulting in information loss. Consequently, different logit patterns may collapse into similar probability distributions, limiting the effectiveness of OOD detection. A natural question arises: *How can we more effectively leverage the discriminative information embedded in non-maximum logits to enhance ID-OOD separability?*

To answer the above question, we first systematically examine the logit distributions of ID and OOD samples. We observe a consistent phenomenon: the relationship between the maximum logit and the remaining logits displays fundamentally different characteristics for ID and OOD data. As shown in Figure 2, ID samples generally have higher maximum logits accompanied by lower non-maximum logits compared to OOD samples. This results in a pronounced "logit gap", defined as the numerical difference between the maximum and the remaining logits, which is consistently and significantly larger for ID samples than for OOD samples. This observed pattern suggests that the relative *logit gap* can naturally serve as a discriminative criteria for effective OOD detection.

Motivated by this observation, we propose **LogitGap**, a novel post-hoc OOD detection approach that effectively leverages the logit-gap to distinguish between ID and OOD samples. By explicitly leveraging the entire logit space, our proposed method significantly enhances the separability between ID and OOD samples (as shown in Figure 1). To further improve its effectiveness, we refine LogitGap by focusing on a more compact and informative subset of the logit space. This improvement is inspired by our empirical observation: the tail region (logits from least likely classes) exhibits substantial overlap between ID and OOD samples, which brings irrelevant and noisy signal. To mitigate this issue, we constrain LogitGap's computation to the top-$N$ logits, naturally excluding the noisy tail logits from the OOD score. To balance information retention (larger $N$) and noise suppression (smaller $N$), we propose a simple yet effective strategy for selecting $N$: choose the value that maximizes the difference between the mean top-$N$ logits of ID and OOD samples, using only a small number of ID samples.

We conduct experiments across diverse OOD scenarios to evaluate the effectiveness of our LogitGap method. Results show that LogitGap outperforms existing OOD scoring functions in both zero-shot and few-shot OOD detection tasks based on the CLIP model. Furthermore, when combined with training-based OOD detection methods, LogitGap brings notable performance gains, highlighting its complementary nature. To assess the generality of our approach, we also evaluate it under traditional OOD settings, where models are trained from scratch using cross-entropy loss on ID data. Across all scenarios, LogitGap consistently achieves significant performance gains, validating its versatility and robustness in various OOD detection scenarios.

Our main contributions can be summarized as follows:

- We propose a novel post-hoc OOD detection method for a pre-trained model, which leverages the information from entire logit space to enhance ID-OOD separability.

- We analyze the limitations of existing OOD scoring methods and theoretically derive their relationship to our LogitGap method.

- We conduct experiments to demonstrate the versatility and effectiveness of proposed Logit-Gap method in various OOD scenarios.

## 2  Related Work

**OOD Detection for Models Trained with Cross-Entropy Loss.**  OOD detection for classification models trained with cross-entropy loss typically falls into two categories. The first involves training-time strategies that endow models with OOD awareness, often requiring access to training data [20, 11, 18]. For example, VOS [11] synthesizes virtual outliers, LogitNorm [52] applies logit normalization, and MOS [22] partitions the class space hierarchically. Other approaches leverage auxiliary OOD samples, including OE [18], which encourages flat predictions, MCD [56], which amplifies entropy discrepancies, UDG [53], which clusters out mixed ID data, and MixOE [57], which mixes ID and OOD data to expand coverage. A key challenge for these methods lies in selecting appropriate auxiliary data while mitigating the risk of overfitting. Another major line of work focuses on post-hoc methods, which are training-free and computationally efficient. These approaches can be broadly categorized into logits-based and feature-based methods. Logits-based approaches derive OOD scores from model outputs, such as MaxLogit [14], ODIN [27], Energy [30], GradNorm [21], and MaxCosine [59]. In contrast, feature-based methods analyze internal representations, including Mahalanobis [25], KNN [45], FDBD [28], ViM [50].

**OOD Detection for Contrastive Learning Pretrained Models.**  For contrastive learning models, MCM [33] extends MSP [17] to vision-language models, while GL-MCM [35] incorporates local feature cues to enhance detection performance. Recent works have increasingly focused on teaching models to recognize "what an input is not". For example, CLIPN [51] fine-tunes CLIP to generate negative prompts corresponding to absent concepts. With the rise of prompt engineering, many approaches improve OOD detection by fine-tuning prompts with a few ID samples. For example, LoCoOp [34] builds on CoOp's [62] by applying OOD regularization through local OOD features. Approaches such as LSN [37] and NegPrompt [26] jointly learn positive and negative prompts to better capture OOD characteristics. ID-like [1] introduces prompts aligned with ID-like regions near the OOD boundary to refine detection through targeted fine-tuning. Additionally, some CLIP-based approaches leverage external concept vocabularies [23, 4] or language models [3] to map OOD samples to auxiliary semantic categories, but such methods are generally constrained to semantic-shift scenarios. While CLIP-based OOD detection methods have demonstrated strong performance, most depend on additional data or external models, limiting their scalability and general applicability. In this work, we propose a lightweight, post-hoc OOD method for pre-trained models, which does not require additional supervision.

## 3  Preliminary

### 3.1  Problem Setting

We study the zero-shot and few-shot OOD detection problem with a pre-trained model $f$. Let $\mathcal{Y}_{in}$ denote the set of *known* classes (in-distribution, ID) for a given classification task. Given a test sample $\boldsymbol{x}$ drawn from the input space $\mathcal{X}$, if its true label $y \notin \mathcal{Y}_{in}$, then $\boldsymbol{x}$ is considered as an OOD sample. The goal of zero-shot OOD detection is to identify samples that do not belong to any of known ID classes, without requiring additional training or exposure to OOD data during model development. Similarly, few-shot OOD detection aims to achieve the same goal, but with access to a small number of training samples from the ID class set $\mathcal{Y}_{in}$.

Formally, in the zero-shot and few-shot OOD problem, the decision function $D : \mathcal{X} \rightarrow \{\text{ID}, \text{OOD}\}$ is typically constructed as:

$$D\left(\boldsymbol{x}\right) = \begin{cases} \text{ID}, & \text{if } S\left(\boldsymbol{x}; f\right) \geq \lambda \\ \text{OOD}, & \text{if } S\left(\boldsymbol{x}; f\right) < \lambda\text{'} \end{cases} \tag{1}$$

where $S : \mathcal{X} \rightarrow \mathbb{R}$ is a scoring function based on pre-trained model $f$, which assigns a scalar confidence score to each input sample. The decision threshold $\lambda$ establishes the decision boundary: samples satisfying $S\left(\boldsymbol{x}\right) < \lambda$ are identified as OOD, while those with $S\left(\boldsymbol{x}\right) \geq \lambda$ are considered ID.

### 3.2  OOD Detection Scoring Function

The design of the scoring function is crucial, as it directly determines the effectiveness of OOD detection. In this subsection, we revisit the design principles of existing logit-based scoring functions,

with a focus on two representative approaches: Maximum Logit (MaxLogit) [14] and Maximum Concept Matching (MCM) [33]. The most straightforward approach, MaxLogit [14], uses the maximum logit value as the confidence score. Formally, given $K$ known classes (*i.e.*, $|\mathcal{Y}_{in}| = K$), for an input sample $\boldsymbol{x}$, the pre-trained model $f$ produces a logit vector as $\boldsymbol{z} = f(\boldsymbol{x}; \mathcal{Y}_{in}) \in \mathbb{R}^K$. The OOD score by MaxLogit is then calculated as:

$$S_{\text{MaxLogit}}(\boldsymbol{x}; f) = \max_k z_k, \tag{2}$$

where $z_k$ denotes the predicted logit for class $k$. However, MaxLogit focus solely on the most confident prediction, disregarding potentially useful information from other classes. A more widely used scoring function is MCM [33], which applies softmax normalization to the logits and takes the maximum probability as the OOD score:

$$S_{\text{MCM}}(\boldsymbol{x}; f) = \max_k \frac{e^{z_k/\tau}}{\sum_{j=1}^K e^{z_j/\tau}}, \tag{3}$$

where $\tau$ is a temperature scaling parameter. Unlike MaxLogit, MCM leverages the full logit distribution through the denominator of the softmax operation. As a result, MCM generally achieves superior separability between ID and OOD samples compared to MaxLogit.

## 4 Method

### 4.1 OOD score: LogitGap

We observe an interesting phenomenon in real-world data: While the mean logit values of ID and OOD samples are often similar, their logit distributions exhibit systematically distinct patterns, as shown in Figure 2. Specifically, ID samples (red line in Figure 2) tend to produce **sharper, more peaked** logit distributions, typically with one dominant logit value corresponding to the predicted class. In contrast, OOD samples often yield **flatter, more uniform** logit distributions, with less pronounced maxima and elevated non-maximum logits. This is quantitatively reflected as: (i) Higher maximum logit values for ID samples compared to OOD samples; (ii) Higher non-predicted class logits in OOD samples relative to ID samples.

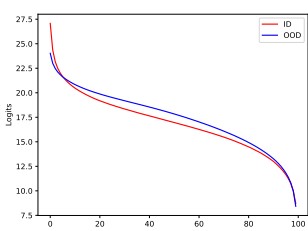

Figure 2: Descending-sorted logits on CLIP ViT-B/16 with ImageNet100 (ID) and iNaturalist (OOD).

Building upon these observations, we propose a simple yet effective OOD scoring function, call LogitGap, which exploits the inherent distributional differences between ID and OOD logits. The core idea is to quantify the average gap between the maximum logit and the remaining logits—a measure that naturally amplifies the contrast between ID and OOD samples. This formulation effectively captures the distinctive peakedness of ID logit predictions, while highlighting the relative flatness of OOD logit distributions, thereby enhancing ID-OOD separability.

Formally, given a logit vector $\boldsymbol{z}$ predicted for a test sample $\boldsymbol{x}$, we first sort its elements in descending order to obtain $\boldsymbol{z}'$, where $z'_n$ denotes the $n$-th largest logit. The LogitGap score is then defined as:

$$S_{\text{LogitGap}}(\boldsymbol{x}; f) = \frac{1}{K-1} \sum_{j=2}^K (z'_1 - z'_j) \tag{4}$$

The main difference between MCM [33] and our proposed LogitGap lies in how they utilize non-maximum class information. MCM implicitly incorporates this information through the normalization process in the softmax denominator, while LogitGap explicitly quantifies the average gap between the maximum logit and all other logits. This explicit formulation allows for more direct utilization of the discriminative patterns observed in logit distributions. To establish a theoretical connection between MCM and LogitGap, we present the following analysis.

**Theorem 4.1.** *Given a $K$-way classification task, let the predicted logit vector for a sample $\boldsymbol{x}$ be $\boldsymbol{z} = [z_1, z_2, \ldots, z_K]$. Let $\boldsymbol{z}' = [z'_1, z'_2, \ldots, z'_K]$ denote the sorted logits in descending order, such*

that $z_1' = \max_k z_k$. Then, for the temperature scaling parameter $\tau$ used in the softmax function, if $\tau > 2(K-1)$, we have

$$\mathrm{FPR}_{\mathrm{LogitGap}}\left(\lambda_L\right) \leq \mathrm{FPR}_{\mathrm{MCM}}\left(\tau, \lambda_{MCM}\right),$$

where $\mathrm{FPR}_{\mathrm{LogitGap}}\left(\lambda_L\right)$ is the false positive rate based on LogitGap with threshold $\lambda_L$. Similarly, $\mathrm{FPR}_{\mathrm{MCM}}\left(\tau, \lambda_{MCM}\right)$ is the false positive rate based on MCM with temperature $\tau$ and threshold $\lambda_{MCM}$.

For clarity, we provide a brief proof sketch below (complete proof is provided in Appendix A).

*proof sketch.* False Positive Rate (FPR) measures the likelihood of an ID sample being misclassified as OOD. A lower FPR indicates better OOD detection performance. Let $Q_x$ denotes the out-of-distribution $P_{x|\mathrm{OOD}}$. By definition, we express the FPR of MCM, denoted as $\mathrm{FPR}_{\mathrm{MCM}}(\tau, \lambda_{\mathrm{MCM}})$, in the following:

$$FPR_{\mathrm{MCM}}(\tau, \lambda_{\mathrm{MCM}}) = Q_x \left( \frac{e^{z_1'/\tau}}{\sum_{j=1}^{k} e^{z_j'/\tau}} > \lambda_{\mathrm{MCM}} \right) \tag{5}$$

Similarly, the FPR of LogitGap can be written as

$$FPR_{\mathrm{LogitGap}}(\tau, \lambda_{\mathrm{L}}) = Q_x \left( \frac{\sum_{j=1}^{K}\left(z_1' - z_j'\right)}{\tau K} > \frac{\lambda_L}{\tau} \right) \tag{6}$$

By introducing an intermediate OOD score function, LogitGap with softmax (LogitGap_softmax), we establish a connection between MCM and LogitGap. We define it as $S_{\mathrm{LogitGap\_softmax}}(\boldsymbol{x}; f) = \frac{\sum_{i=1}^{K}[e^{z_1'/\tau} - e^{z_i'/\tau}]}{K\sum_{j=1}^{K} e^{z_j'/\tau}}$. Accordingly, we express the FPR of LogitGap_softmax as

$$FPR_{\mathrm{LogitGap-softmax}}(\tau, \lambda_{\mathrm{LM}}) = Q_x \left( \frac{\sum_{i=1}^{K}[e^{z_1'/\tau} - e^{z_i'/\tau}]}{K\sum_{j=1}^{K} e^{z_j'/\tau}} > \lambda_{\mathrm{LM}} \right), \tag{7}$$

where $\lambda_{\mathrm{MCM}} = \lambda_{\mathrm{LM}} + \frac{1}{K}$.

Then, we aim to demonstrate the performance of LogitGap is guaranteed to surpass that of MCM, which is equal to finding the conditions under which the inequality holds: $FPR_{LogitGap}(\tau, \lambda_{\mathrm{L}}) < FPR_{MCM}(\tau, \lambda_{\mathrm{MCM}})$. By combining the results from Eq.(5) Eq.(6), we can derive that the inequality holds with the condition of $\tau > 2(K-1)$.

$\square$

Theorem 4.1 demonstrates that when the temperature scaling parameter $\tau$ exceeds a certain threshold, the OOD detection performance of LogitGap is guaranteed to surpass that of MCM [33]. This result stems from fundamental differences in how the two methods process logit information: (i) **Information Preservation.** MCM operates on softmax-normalized probabilities, which inherently compress logit magnitudes. Differently, LogitGap directly exploits the raw logit margins, preserving richer discriminative cues. (ii) **Temperature Sensitivity.** The performance gap increases with higher $\tau$: for MCM: higher temperatures exacerbate information loss by overly dispersing the probability mass; For LogitGap, the margin-based formulation remains robust regardless of $\tau$.

## 4.2  Logits Selection for Focused Scoring

While LogitGap utilize the complete set of $K$ predicted logits, we identify a critical limitation in $K$-way classification: certain classes consistently yield negligible activations, particularly those that are semantically or visually unrelated to the input. These inactive logits provide limited discriminative information and thus contribute little to OOD detection. The issue becomes more pronounced as $K$ increases, since a larger proportion of irrelevant logits is introduced. These irrelevant logits add noise, reducing the clarity of the distributional differences between ID and OOD samples.

By eliminating redundant tail information, the structural differences between ID and OOD prediction distributions become more pronounced and easier to exploit. To achieve this, we propose narrowing the focus to a more compact and informative subset of the logit space. Formally, we sort the predicted

logits in descending order and select the top $N$ (out of $K$) logits, resulting in a reduced prediction vector. We then apply the LogitGap scoring function to this truncated logit vector

$$S_{\text{LogitGap-topN}}\left(\boldsymbol{x}; f\right) = \frac{1}{N-1}\sum_{j=2}^{N}\left(z_1' - z_j'\right) \tag{8}$$

**How to Determine the Hyperparameter $N$.** The effectiveness of LogitGap largely depends on the choice of the hyperparameter $N$, which controls the number of top logits retained for OOD scoring. An appropriate selection of $N$ strikes a balance between capturing informative signals and avoiding noisy, uninformative logits. Formally, given the sorted logits $\boldsymbol{z}'$, the LogitGap-topN score can be equivalently reformulated as:

$$S_{\text{LogitGap-topN}}\left(\boldsymbol{x}; f\right) = \frac{1}{N-1}\sum_{j=2}^{N}\left(z_1' - z_j'\right) = \frac{1}{N-1}\sum_{j=2}^{N}z_1' - \frac{1}{N-1}\sum_{j=2}^{N}z_j' = z_1' - \bar{z}_N' \tag{9}$$

where $\bar{z}_N'$ denotes the mean of the logits ranked from second to the $N$-th largest. Then, for OOD samples drawn from $P_{\text{OOD}}\left(\boldsymbol{x}\right)$ and ID samples drawn from $P_{\text{ID}}\left(\boldsymbol{x}\right)$, our objective is to identify the optimal value of $N$ that maximizes the score discrepancy between them. This can be expressed as:

$$\arg\max_N \left(\mathbb{E}_{\boldsymbol{x}\sim P_{\text{ID}}}\left[S_{\text{LogitGap-topN}}\left(\boldsymbol{x}\right)\right] - \mathbb{E}_{\boldsymbol{x}\sim P_{\text{OOD}}}\left[S_{\text{LogitGap-topN}}\left(\boldsymbol{x}\right)\right]\right)$$
$$= \arg\max_N \left(\mathbb{E}_{\boldsymbol{x}\sim P_{\text{ID}}}\left[z_1' - \bar{z}_N'\right] - \mathbb{E}_{\boldsymbol{x}\sim P_{\text{OOD}}}\left[z_1' - \bar{z}_N'\right]\right)$$
$$= \arg\max_N \left(\mathbb{E}_{\boldsymbol{x}\sim P_{\text{ID}}}\left[z_1'\right] - \mathbb{E}_{\boldsymbol{x}\sim P_{\text{OOD}}}\left[z_1'\right] + \mathbb{E}_{\boldsymbol{x}\sim P_{\text{OOD}}}[\bar{z}_N'] - \mathbb{E}_{\boldsymbol{x}\sim P_{\text{ID}}}[\bar{z}_N']\right)$$
$$= \arg\max_N \left(\mathbb{E}_{\boldsymbol{x}\sim P_{\text{OOD}}}[\bar{z}_N'] - \mathbb{E}_{\boldsymbol{x}\sim P_{\text{ID}}}[\bar{z}_N']\right), \tag{10}$$

Equation (10) provides a criterion for selecting the hyperparameter $N$. However, in practical scenarios, it is often challenging to obtain authentic OOD samples. To address this, we assume access to a small ID validation set (containing no more than 100 samples) and simulate potential OOD data by applying interpolation-based transformations to the ID features and injecting random noise. Based on these synthesized samples, we can estimate an appropriate value for $N$ through Equation (10).

## 5  Experiments

### 5.1  Experimental Settings

**Datasets.** We evaluate the effectiveness of LogitGap across multiple OOD detection benchmarks. Specifically, we use ImageNet [7] or ImageNet-100 as ID datasets, while NINCO [2], ImageNet-OOD [55], and ImageNet-O [19] are used as OOD datasets. Each OOD dataset contains categories that are disjoint from those in the corresponding ID dataset. Following MCM [33], we also construct a more challenging OOD detection task by alternately using ImageNet-10 and ImageNet-20 as the ID and OOD datasets. This setting leverages semantic similarity within the label space to increase task difficulty, providing a more rigorous evaluation of OOD separability.

**Comparison Methods.** We adopt CLIP [39] with ViT-B/16 [10] backbone as the pre-trained model for zero-shot OOD detection. Our evaluation encompasses both zero-shot and few-shot settings. For the few-shot OOD detection setting, we randomly select one (one-shot) or four (four-shot) samples from each ID class and use these limited samples to fine-tune CLIP. To provide a comprehensive comparison, we benchmark against several representative post-hoc OOD detection methods, including MCM [33], MaxLogit [14], Energy [30] and GL-MCM [35], as well as an enhancing method TAG [31]. Additionally, we evaluate two few-shot OOD approaches: CoOp [62] and ID-Like [1]. Specifically, CoOp [62] fine-tunes prompts using limited labeled samples, serving as a simple few-shot baseline. ID-Like [1] jointly optimizes ID-like prompts and synthesizes outliers from semantically related ID data, thereby improving detection of semantically similar OOD samples. We compare these methods against two variants of our approach: **LogitGap**, which uses a fixed value of $N$ set to 20% of the total number of classes, and **LogitGap\***, where $N$ is adaptively determined using an ID validation set along with synthetic OOD data.

**Evaluation Metrics.** We use three standard metrics commonly used in OOD detection literature: (1) False Positive Rate (FPR95): Measures the probability that an OOD sample is misclassified as ID

Table 1: OOD Detection performance on CLIP ViT-B/16 under zero-shot and few-shot settings. Results are reported with ImageNet as the ID dataset in the semantic shift scenario.

| | OOD Dataset | | | | | | | | | AVG | | |
| | NINCO | | | ImageNet-O | | | ImageNetOOD | | | | | |
| Method | FPR95 ↓ | AUROC ↑ | AUPR ↑ | FPR95 ↓ | AUROC ↑ | AUPR ↑ | FPR95 ↓ | AUROC ↑ | AUPR ↑ | FPR95 ↓ | AUROC ↑ | AUPR ↑ |
|---|---|---|---|---|---|---|---|---|---|---|---|---|
| | | | | | | Zero-shot | | | | | | |
| Energy [30] | 84.11 | 72.04 | 95.65 | 81.60 | 75.58 | 98.66 | 79.12 | 76.77 | 83.96 | 81.61 | 74.8 | 92.76 |
| +TAG [31] | 83.11 | 71.15 | 95.46 | 79.65 | 77.10 | 98.79 | 78.34 | 78.03 | 85.81 | 80.37 | 75.43 | 93.35 |
| MCM [33] | 79.67 | 73.59 | 95.87 | 75.85 | 79.52 | 98.93 | 80.98 | 78.33 | 85.42 | 78.83 | 77.15 | 93.41 |
| +TAG [31] | 81.63 | 71.33 | 95.41 | 77.60 | 79.95 | 98.97 | 82.99 | 78.79 | 86.42 | 80.74 | 76.69 | 93.6 |
| MaxLogit [14] | 79.41 | 74.35 | 96.03 | 77.15 | 77.85 | 98.79 | 75.85 | 78.67 | 85.16 | 77.47 | 76.96 | 93.33 |
| +TAG [31] | 77.70 | 73.92 | 95.95 | 74.50 | 79.48 | 98.92 | **75.17** | 80.07 | **87.01** | 75.79 | 77.82 | **93.96** |
| GL-MCM | **74.38** | 76.03 | 96.26 | 72.35 | 79.50 | 98.88 | 79.16 | 77.31 | 83.67 | 75.30 | 74.74 | 86.23 |
| LogitGap | 76.83 | 76.43 | 96.37 | 72.35 | 81.32 | **99.03** | 76.37 | 79.95 | 86.06 | 75.18 | 79.23 | 93.82 |
| LogitGap* | 77.42 | **76.51** | **96.38** | 71.95 | **81.45** | **99.03** | 75.40 | **80.27** | 86.21 | **74.92** | **79.41** | 93.87 |
| | | | | | | One-shot | | | | | | |
| CoOp [62] | 84.01 | 68.83 | 94.91 | 75.55 | 78.63 | 98.84 | 82.25 | 76.88 | 84.20 | 80.60 | 74.78 | 92.65 |
| +LogitGap* | **82.85** | **73.57** | **95.91** | **74.85** | **79.40** | **98.88** | **78.31** | **78.09** | **84.55** | **78.67** | **77.02** | **93.11** |
| ID-Like [1] | 73.02 | 75.83 | 95.86 | 80.95 | 69.81 | 98.09 | 83.23 | 68.57 | 75.22 | 79.07 | 71.40 | 89.72 |
| +LogitGap* | **68.78** | **80.90** | **97.07** | **71.20** | **78.21** | **98.76** | **75.06** | **76.33** | **82.40** | **71.68** | **78.48** | **92.74** |
| | | | | | | Four-shot | | | | | | |
| CoOp [62] | 81.46 | 70.16 | 95.14 | 75.40 | 79.80 | 98.93 | 80.40 | 78.56 | 85.56 | 79.09 | 76.17 | 93.21 |
| +LogitGap* | **80.69** | **73.97** | **95.95** | **72.70** | **81.10** | **99.01** | **76.07** | **80.16** | **86.42** | **76.49** | **78.41** | **93.79** |
| ID-Like [1] | 81.15 | 71.79 | 95.01 | 82.50 | 68.41 | 97.92 | 88.54 | 62.40 | 70.25 | 84.06 | 67.53 | 87.73 |
| +LogitGap* | **77.51** | **75.39** | **95.91** | **78.10** | **75.29** | **98.57** | **82.65** | **72.88** | **79.85** | **79.42** | **74.52** | **91.44** |

when the true positive rate of ID samples is fixed at $95\%$. (2) Area Under the Receiver Operating Characteristic Curve (AUROC), and (3) Area Under the Precision-Recall Curve (AUPR).

## 5.2 Results

**Zero-Shot OOD Detection.** We first conduct a comprehensive evaluation of various methods in the zero-shot OOD detection setting. Our experimental setup utilizes ImageNet and ImageNet-100 as ID datasets, comparing multiple OOD scoring functions against our proposed LogitGap approach. As shown in Table 1 and Table 2, both LogitGap and its adaptive variant LogitGap* achieve superior performance compared to existing methods. Specifically, on ImageNet as ID dataset, LogitGap reduces FPR95 by $3.65\%$ compared to MCM [33]. On ImageNet-100, the improvement reaches $5.78\%$. These significant performance gains highlight the effectiveness of our logit-gap based scoring approach. Furthermore, LogitGap* typically provides additional improvements over the fixed-parameter LogitGap, validating the advantage of our adaptive parameter selection strategy[3].

**Few-Shot OOD Detection.** We further investigate the integration of LogitGap with several recent few-shot OOD detection approaches, including CoOp [62] and ID-Like [1]. Specifically, CoOp [62] enhances in-distribution classification by prompt tuning, and ID-Like [1] extends this framework by incorporating negative prompts to enhance OOD detection. Following [1], we conduct rigorous experiments under both one-shot and four-shot finetuning configurations. The results are summarized in Table 1 and Table 2. As shown, LogitGap* consistently enhances the performance of both baseline methods across all evaluation metrics. For instance, using ImageNet as ID dataset, integrating LogitGap with ID-Like [1] improves FPR95 and AUROC by $7.39\%$ and $7.08\%$ under the 1-shot setting, respectively (Table 1). These significant performance gains demonstrate that LogitGap can serve as an effective complementary module to existing OOD detection frameworks, effectively amplifying their OOD detection capabilities.

**Hard OOD Detection.** To thoroughly evaluate the robustness of LogitGap, we further conduct specialized experiments focusing on two particularly challenging categories of hard OOD samples: Semantically Hard OOD Detection and Covariate Shifts OOD Detection.

**1. Semantically Hard OOD Detection.** Prior work [33] has demonstrated that OOD detection becomes particularly challenging when OOD samples exhibit semantic similarity to ID data. To evaluate this difficult scenario, we adopt the experimental setup from MCM [33], alternately

---

[3]In Table 2, the performances of LogitGap and LogitGap* are identical because the parameter search result is equal to 20% of the number of categories.

Table 2: OOD Detection performance on CLIP ViT-B/16 under zero-shot and few-shot settings. Results are reported with ImageNet-100 as the ID dataset in the semantic shift scenario.

| | NINCO | | | OOD Dataset ImageNet-O | | | ImageNetOOD | | | AVG | | |
|---|---|---|---|---|---|---|---|---|---|---|---|---|
| Method | FPR95 ↓ | AUROC ↑ | AUPR ↑ | FPR95 ↓ | AUROC ↑ | AUPR ↑ | FPR95 ↓ | AUROC ↑ | AUPR ↑ | FPR95 ↓ | AUROC ↑ | AUPR ↑ |
| | | | | | | Zero-shot | | | | | | |
| Energy [30] | 52.08 | 88.97 | 88.36 | 54.05 | 89.01 | 95.29 | 51.25 | 89.44 | 68.42 | 52.46 | 89.14 | 84.02 |
| MCM [33] | 50.03 | 89.82 | 89.28 | 48.55 | 90.93 | 96.28 | 51.25 | 90.29 | 70.17 | 49.94 | 90.35 | 85.24 |
| MaxLogit [14] | 45.51 | 89.79 | 88.98 | 47.95 | 90.13 | 95.74 | 46.57 | 90.37 | 70.00 | 46.68 | 90.10 | 84.91 |
| LogitGap | **42.62** | **91.08** | **90.55** | **43.65** | **92.02** | **96.74** | **46.21** | **91.21** | **71.59** | **44.16** | **91.44** | **86.29** |
| LogitGap* | **42.62** | **91.08** | **90.55** | **43.65** | **92.02** | **96.74** | **46.21** | **91.21** | **71.59** | **44.16** | **91.44** | **86.29** |
| | | | | | | One-shot | | | | | | |
| CoOp [62] | 58.03 | 89.30 | 89.46 | 52.85 | 89.61 | 95.55 | 62.78 | 87.06 | 60.96 | 57.89 | 88.66 | 81.99 |
| +LogitGap* | **51.28** | **90.64** | **90.87** | **50.75** | **90.32** | **95.93** | **60.57** | **87.77** | **62.12** | **54.20** | **89.58** | **82.97** |
| ID-Like [1] | 58.56 | 85.76 | 83.65 | 61.80 | 84.34 | 92.36 | 75.00 | 79.93 | 44.65 | 65.12 | 83.34 | 73.55 |
| +LogitGap* | **39.93** | **91.44** | **90.70** | **40.25** | **90.65** | **95.48** | **45.73** | **89.89** | **66.06** | **41.97** | **90.66** | **84.08** |
| | | | | | | Four-shot | | | | | | |
| CoOp [62] | 60.43 | 85.49 | 83.68 | 51.15 | 90.05 | 95.81 | 59.56 | 88.39 | 64.25 | 57.05 | 87.98 | 81.25 |
| +LogitGap* | **52.84** | **87.67** | **86.02** | **47.05** | **91.14** | **96.30** | **53.49** | **89.72** | **66.91** | **51.13** | **89.51** | **83.08** |
| ID-Like [1] | 46.65 | 90.74 | 90.14 | 58.25 | 86.51 | 93.84 | 71.82 | 81.22 | 47.06 | 58.91 | 86.16 | 77.01 |
| +LogitGap* | **31.81** | **93.44** | **92.92** | **44.80** | **90.44** | **95.72** | **54.68** | **88.57** | **65.50** | **43.76** | **90.82** | **84.71** |

Table 3: OOD Detection performance on CLIP ViT-B/16 in the semantically hard scenario. "ImageNet-10 / ImageNet-20": ImageNet-10 as ID and ImageNet-20 as OOD.

| | ImageNet-10 / ImageNet-20 | | | ImageNet-20 / ImageNet-10 | | |
|---|---|---|---|---|---|---|
| Method | FPR95 ↓ | AUROC ↑ | AUPR ↑ | FPR95 ↓ | AUROC ↑ | AUPR ↑ |
| Energy [30] | 8.70 | 98.45 | 97.41 | 20.80 | 96.72 | 98.58 |
| MCM [33] | 6.40 | 98.88 | 98.23 | 22.60 | 97.17 | 98.72 |
| MaxLogit [14] | 8.40 | 98.54 | 97.55 | 20.20 | 97.25 | 98.79 |
| LogitGap | 3.80 | 99.05 | 98.52 | **14.00** | 98.14 | **99.15** |
| LogitGap* | **3.00** | **99.10** | **98.60** | 14.20 | **98.15** | **99.15** |

using ImageNet-10 and ImageNet-20 as ID and OOD datasets. As shown in Table 3, LogitGap demonstrates superior performance across all evaluation metrics compared to existing methods. Specifically, when detecting ImageNet-20 as OOD against ImageNet-10 ID, LogitGap reduces FPR95 by 3.40% compared to MCM [33]. In the reverse configuration (ImageNet-20 as ID vs. ImageNet-10 as OOD), the improvement reaches 8.40%. These significant margins highlight LogitGap's enhanced capability to distinguish semantically related distributions, addressing a key challenge in real-world OOD detection scenarios.

**2. Covariate Shifts OOD Detection.** While CLIP exhibits strong generalization, its performance degrades significantly under covariate shifts. For instance, fine-tuning CLIP on ImageNet leads to notable drops on covariate-shifted variants like ImageNet-Sketch [49] and ImageNet-A [19], where the visual appearance of test samples significantly differs from that of ImageNet [62]. To assess LogitGap under hard OOD conditions, we consider a covariate shift setting. Specifically, we construct ID data using limited samples (one-shot or four-shot) from ImageNet, while treating ImageNet-R [16], ImageNet-A, and ImageNet-Sketch as OOD datasets. LogitGap is integrated into several recent few-shot OOD detection methods. The results are summarized in Table 4.

From Table 4, we observe that: (1) Increasing ID training samples does not consistently enhance OOD detection performance in some few-shot OOD methods. For instance, ID-like [1] shows higher FPR95 in 4-shot versus 1-shot training. We hypothesize that this phenomenon arises from the strong generalization capability of fine-tuned CLIP models: while aiding ID classification, it may also lead to overconfident prediction, thereby misclassifying OOD samples as incorrect ID classes and ultimately impairing OOD detection. (2) The proposed LogitGap method consistently enhances OOD detection performance across various covariate shift scenarios and different few-shot OOD methods. For instance, LogitGap reduces FPR95 by 2.20% (1-shot) and 1.17% (4-shot) for ID-Like [1], underscoring its effectiveness under varying few-shot conditions.

Table 4: OOD Detection performance on CLIP ViT-B/16 under few-shot setting. Results are reported with ImageNet as the ID dataset in the covariate shift scenario.

| Method | ImageNet-R FPR95 ↓ | AUROC ↑ | AUPR ↑ | ImageNet-A FPR95 ↓ | AUROC ↑ | AUPR ↑ | ImageNet-Sketch FPR95 ↓ | AUROC ↑ | AUPR ↑ | AVG FPR95 ↓ | AUROC ↑ | AUPR ↑ |
|---|---|---|---|---|---|---|---|---|---|---|---|---|
| | | | | | | One-shot | | | | | | |
| CoOp [62] | 80.18 | 72.44 | **91.54** | 77.75 | 77.40 | 95.65 | 85.53 | 67.21 | **67.06** | 81.15 | 72.35 | **84.75** |
| +LogitGap* | **79.97** | **71.14** | 90.81 | **75.72** | **79.77** | 96.25 | **82.81** | **68.37** | 67.00 | **79.50** | **73.09** | 84.69 |
| ID-Like [1] | 79.68 | 75.51 | 92.61 | 45.67 | 88.65 | 97.77 | 78.14 | 75.31 | 75.32 | 67.83 | 79.82 | 88.57 |
| +LogitGap* | **76.11** | **77.28** | **93.20** | **43.36** | **89.99** | **98.14** | **77.43** | **76.06** | **76.42** | **65.63** | **81.11** | **89.25** |
| | | | | | | Four-shot | | | | | | |
| CoOp [62] | **78.38** | 73.53 | **92.02** | 74.87 | 79.20 | 96.06 | 84.68 | 68.54 | **69.06** | 79.31 | 73.76 | **85.71** |
| +LogitGap* | 79.59 | 71.94 | 91.29 | **74.20** | **80.62** | 96.45 | **82.68** | **69.23** | 68.91 | **78.82** | **73.93** | 85.55 |
| ID-Like [1] | 85.69 | 71.93 | 91.39 | 61.27 | 84.80 | 97.06 | 87.07 | 67.45 | 67.47 | 78.01 | 74.73 | 85.31 |
| +LogitGap* | **84.45** | 71.16 | 91.03 | **59.13** | **86.31** | **97.44** | 86.95 | 66.80 | 67.47 | **76.84** | **74.76** | 85.31 |

## 5.3 Further Discussion

**Influence of the Hyperparameter $N$ on LogitGap Performance.** We examine the impact of the hyperparameter $N$ on the performance of the LogitGap method. We conduct experiments by utilizing ImageNet-100 as the ID dataset in the semantic shift sanario. As illustrated in Figure 3, the FPR95 performance on all three OOD datasets degrades when $N$ is either too small or too large. This trend can be attributed to the trade-off in the representation capacity of the logits space. When $N$ is too small, the logits space is overly compact and lacks sufficient discriminative information, thereby limiting the effectiveness of LogitGap. In contrast, an excessively large $N$ introduces a substantial amount of tail information, where the distinction between ID and OOD samples becomes less pronounced. This weakens the discriminative signal and reduces overall performance. Importantly, we find that LogitGap exhibits relatively stable and consistently strong performance when $N$ is set between 20% and 50% of the total number of classes. This observation suggests that a moderately compact logits space provides a favorable balance between retaining information and suppressing noise, offering a robust choice for selecting $N$ in real-world OOD detection scenarios.

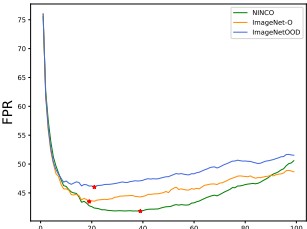

Figure 3: Variation of FPR95 performance with increasing $N$. Results are reported on CLIP ViT-B/16 with ImageNet-100 as the ID dataset in semantic shift sanario.

**Logit Distribution Differences Between ID and OOD Data.** The design of LogitGap is motivated by the observation that the logit distributions of ID and OOD samples exhibit distinct patterns: ID samples tend to have higher maximum logit, whereas OOD samples display higher logits among the non-predicted classes. To verify the generality of this phenomenon, we analyze the logit distribution statistics using ImageNet as the ID dataset on CLIP ViT-L/14. Specifically, $z'_1$ denotes the average maximum logit, while $\bar{z}'_N$ represents the average logit over all non-predicted classes. As shown in Figure 4, ID samples consistently show larger $z'_1$ and smaller $\bar{z}'_N$ than OOD samples, which confirms the broad consistency of this observation.

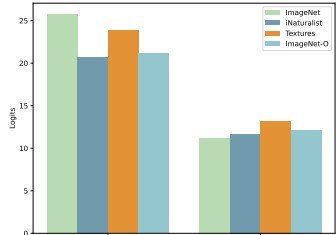

Figure 4: Logit Statistics on ViT-L/14 with ImageNet as ID. $z'_1$ and $\bar{z}'_N$ denote the average maximum logit and the average logit of non-predicted classes, respectively.

**Different Variants of LogitGap.** Motivated by the distinct logit distribution patterns of ID and OOD samples, we design LogitGap, which subtracts the average of non-maximum logits from the maximum logit. Since ID samples usually have higher maximum and lower remaining logits, this formulation effectively enhances ID–OOD separation. In addition, we introduce three variants. LogitGap_exp, LogitGap_square, and LogitGap_sqrt apply exponential, square, square root transformations, respectively, to further transform score differences. As shown in Table 5, these variants perform comparably to the original LogitGap, suggesting that different nonlinear transformations can achieve similarly effective discrimination.

Table 5: Ablation study on different variants of LogitGap. Results are reported on CLIP ViT-B/16 with ImageNet as the ID dataset in zero-shot setting.

| | | | | OOD Dataset | | | | | | AVG | | |
| | NINCO | | | ImageNet-O | | | ImageNetOOD | | | | | |
| Method | FPR95 ↓ | AUROC ↑ | AUPR ↑ | FPR95 ↓ | AUROC ↑ | AUPR ↑ | FPR95 ↓ | AUROC ↑ | AUPR ↑ | FPR95 ↓ | AUROC ↑ | AUPR ↑ |
|---|---|---|---|---|---|---|---|---|---|---|---|---|
| LogitGap | 77.42 | 76.51 | 96.38 | 71.95 | 81.45 | 99.03 | 75.40 | 80.27 | 86.21 | 74.92 | 79.41 | 93.87 |
| LogitGap_exp | 77.15 | 76.67 | 96.42 | 71.40 | 81.54 | 99.04 | 74.33 | 80.55 | 86.51 | 74.48 | 79.59 | 93.99 |
| LogitGap_square | 76.81 | 76.80 | 96.45 | 71.25 | 81.53 | 99.04 | 74.17 | 80.97 | 86.95 | 74.08 | 79.77 | 94.15 |
| LogitGap_sqrt | 77.15 | 76.43 | 96.36 | 71.50 | 81.58 | 99.04 | 75.44 | 80.36 | 86.33 | 74.70 | 79.46 | 93.91 |

Table 6: OOD Detection performance on ResNet-50 using CIFAR-10 as the ID dataset.

| | Near OOD | | | | Far OOD | | | | | | | | AVG | |
| | CIFAR100 | | TIN | | MNIST | | SVHN | | TEXTURE | | PLACES365 | | | |
| Method | FPR95 | AUROC | FPR95 | AUROC | FPR95 | AUROC | FPR95 | AUROC | FPR95 | AUROC | FPR95 | AUROC | FPR95 ↓ | AUROC ↑ |
|---|---|---|---|---|---|---|---|---|---|---|---|---|---|---|
| MSP [17] | 53.10 | 87.19 | 43.25 | 88.87 | 23.63 | 92.63 | 25.80 | 91.46 | 34.96 | 89.89 | 42.26 | 88.92 | 37.17 | 89.83 |
| MaxLogit [14] | 66.60 | 86.31 | 56.06 | 88.72 | 25.06 | 94.15 | 35.09 | 91.69 | 51.71 | 89.41 | 54.84 | 89.14 | 48.23 | 89.90 |
| Energy [30] | 66.59 | 86.36 | 56.08 | 88.8 | 24.99 | 94.32 | 35.15 | 91.78 | 51.84 | 89.47 | 54.86 | 89.25 | 48.25 | 90.00 |
| ODIN [27] | 77.04 | 82.18 | 75.32 | 83.55 | 23.81 | **95.24** | 68.61 | 84.58 | 67.74 | 86.94 | 70.35 | 85.07 | 63.81 | 86.26 |
| GEN [32] | 58.76 | 87.21 | 48.58 | 89.20 | 23.00 | 93.83 | 28.14 | 91.97 | 40.72 | 90.14 | 47.04 | 89.46 | 41.04 | 90.30 |
| Mahalanobis [25] | 52.81 | 83.59 | 47.01 | 84.81 | 27.30 | 90.10 | 25.95 | 91.19 | 27.92 | **92.69** | 47.66 | 84.90 | 38.11 | 87.88 |
| ReAct [44] | 67.40 | 85.93 | 59.71 | 88.29 | 33.76 | 92.81 | 50.23 | 89.12 | 51.40 | 89.38 | 44.21 | 90.35 | 51.12 | 89.31 |
| SHE [58] | 80.99 | 80.31 | 78.27 | 82.76 | 42.23 | 90.43 | 62.75 | 86.37 | 84.59 | 81.57 | 76.35 | 82.89 | 70.86 | 84.06 |
| NCI [29] | 52.46 | 87.84 | 42.92 | 89.50 | 28.94 | 92.08 | 31.70 | 90.67 | **27.59** | 91.97 | **35.65** | **90.36** | 36.54 | 90.40 |
| LogitGap | **49.17** | **88.01** | **39.91** | **89.70** | 22.56 | 93.44 | 25.41 | 92.10 | 33.00 | 90.73 | 39.38 | 89.74 | **34.91** | **90.62** |
| | | | | | | | | | | | | | | |
| OE [18] | 35.84 | 90.81 | **2.50** | **99.25** | 25.89 | 90.63 | 1.10 | **99.69** | 10.61 | 98.01 | 13.48 | 97.14 | 14.90 | 95.92 |
| +LogitGap | **35.18** | **91.18** | 2.81 | 99.21 | **24.96** | **91.52** | 1.18 | 99.65 | **9.76** | **98.10** | **13.20** | **97.18** | **14.52** | **96.14** |
| MixOE [57] | 65.58 | 87.01 | 46.79 | 90.13 | 36.89 | 91.79 | 15.56 | **94.48** | 33.40 | 92.00 | 38.87 | 91.18 | 39.52 | 91.10 |
| +LogitGap | **52.30** | **88.22** | **35.34** | **90.81** | **28.04** | **92.26** | 15.54 | 94.46 | **25.80** | **92.45** | **30.38** | **91.53** | **31.23** | **91.62** |

**Effectiveness on Traditional OOD Detection.** To rigorously assess the generalizability of LogitGap, we extend our evaluation to traditional OOD detection setting. Following the OpenOOD protocol [54], we use CIFAR-10 [24] as ID dataset and adopt the standard OpenOOD benchmark splits for OOD evaluation. The OOD benchmarks include both **near-OOD datasets**: CIFAR-100, Tiny ImageNet, and **far-OOD datasets**: MNIST [8], SVHN [36], Texture [5], and Places365 [61]. In this setup, we employ a ResNet-50 [13] backbone trained from scratch on ID data using cross-entropy loss. We compare LogitGap against a comprehensive suite of baselines: (i) Logit-based post-hoc methods: MSP [17], MaxLogit [14], ODIN [27], Energy [30], GEN [32]; (ii) Internal network statistics-based methods: Mahalanobis [25], ReAct [44], NCI [29], SHE [58]; (iii) Training-based methods with auxiliary data: Outlier Exposure (OE) [18], MixOE [57].

As shown in Table 6, LogitGap consistently outperforms all baselines across benchmarks. We observe that: (1) Compared to the logit-based method MSP [17], LogitGap reduces FPR95 by 2.26%. (2) Unlike NCI [29], which relies on the relationship between features and classifier weights, LogitGap is fully end-to-end without extra feature extraction. (3) When combined with auxiliary-data methods such as OE [18] and MixOE [57], LogitGap further improves their baselines — achieving a notable 8.29% FPR95 reduction on MixOE. These results demonstrate the strong generalization capability of LogitGap across diverse OOD detection paradigms in the traditional setting.

## 6 Conclusion

In this work, we revisit the OOD detection problem from the perspective of logits space and reveal key limitations of existing logits-based scoring methods. To overcome these challenges, we propose LogitGap, a simple yet effective approach that enhances ID-OOD separability by explicitly leveraging the gap between the maximum logit and the remaining logits. Furthermore, we develop a lightweight, data-efficient strategy to automatically determine the optimal subset size for scoring. Extensive experiments show that LogitGap consistently outperforms existing methods across both traditional vision architectures and pre-trained vision-language models, while maintaining ease of deployment.

**Limitations and Broader Impact.** While LogitGap demonstrates strong performance in visual classification tasks, its current design is primarily tailored to visual models. In future work, we plan to extend LogitGap to other modalities, such as text or speech, to enable robust OOD detection across a broader range of input types.

## Acknowledgments

This work is partially supported by National Science and Technology Major Project 2021ZD0111901 and National Natural Science Foundation of China 62376259, 62576334, 62306301.

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

# A  Theoretical Analysis: LogitsGap vs Softmax Scaling

In this section, we provide the proof of Theorem 4.1 from Section 4, which offers a detailed analysis of the relationship between LogitGap and MCM [33]. We first introduce the necessary notation and definitions.

**Notations.** Formally, in the out-of-distribution problem, the decision function $D : \mathcal{X} \to \{\text{ID}, \text{OOD}\}$ is typically constructed as:

$$D(\boldsymbol{x}) = \begin{cases} \text{ID}, & \text{if } S(\boldsymbol{x}; f) \geq \lambda \\ \text{OOD}, & \text{if } S(\boldsymbol{x}; f) < \lambda \end{cases}, \tag{11}$$

where $S : \mathcal{X} \to \mathbb{R}$ is a scoring function based on pre-trained model $f$, which assigns a scalar confidence score to each input sample. The decision threshold $\lambda$ establishes the decision boundary: samples satisfying $S(\boldsymbol{x}) < \lambda$ are identified as OOD, while those with $S(\boldsymbol{x}) \geq \lambda$ are considered ID.

We present the OOD scoring functions for LogitGap and MCM. Given a logit vector $\boldsymbol{z}$ predicted for a test sample $\boldsymbol{x}$, we sort its elements in descending order to obtain $\boldsymbol{z}'$. LogitGap exploits the inherent distributional differences between ID and OOD logits [4]:

$$S_{\text{LogitGap}}(\boldsymbol{x}; f) = \frac{1}{K} \sum_{j=1}^{K} (z_1' - z_j'), \tag{12}$$

where $z_j'$ denotes the $j$-th largest logit.

MCM [33] applies softmax normalization to the logits and takes the maximum probability as the OOD score:

$$S_{\text{MCM}}(\boldsymbol{x}; f) = \max_k \frac{e^{z_k/\tau}}{\sum_{j=1}^{K} e^{z_j/\tau}}, \tag{13}$$

where $\tau$ is a temperature scaling parameter.

Furthermore, we give the OOD decision function based on the corresponding score:

$$D_{\text{LogitGap}}(\boldsymbol{x}) = \begin{cases} \text{ID}, & \text{if } S_{\text{LogitGap}}(\boldsymbol{x}; f) \geq \lambda_{\text{LogitGap}} \\ \text{OOD}, & \text{if } S_{\text{LogitGap}}(\boldsymbol{x}; f) < \lambda_{\text{LogitGap}} \end{cases}, \tag{14}$$

$$D_{\text{MCM}}(\boldsymbol{x}) = \begin{cases} \text{ID}, & \text{if } S_{\text{MCM}}(\boldsymbol{x}; f) \geq \lambda_{\text{MCM}} \\ \text{OOD}, & \text{if } S_{\text{MCM}}(\boldsymbol{x}; f) < \lambda_{\text{MCM}} \end{cases}. \tag{15}$$

**Remarks.** By convention, $\lambda_{\text{LogitGap}}$ and $\lambda_{\text{MCM}}$ are typically chosen such that the true positive rate is at 95%. For brevity, we will use $\lambda_{\text{L}}$ to denote $\lambda_{\text{LogitGap}}$ throughout the rest of this paper.

**Theorem A.1.** *Given a $K$-way classification task, let the predicted logit vector for a sample $\boldsymbol{x}$ be $\boldsymbol{z} = [z_1, z_2, \ldots, z_K]$. Let $\boldsymbol{z}' = [z_1', z_2', \ldots, z_K']$ denote the sorted logits in descending order, such that $z_1' = \max_k z_k$. Then, for the temperature scaling parameter $\tau$ used in the softmax function, if $\tau > 2(K-1)$, we have*

$$\text{FPR}_{\text{LogitGap}}(\lambda_{\text{L}}) \leq \text{FPR}_{\text{MCM}}(\tau, \lambda_{\text{MCM}}),$$

*where $\text{FPR}_{\text{LogitGap}}(\lambda_{\text{L}})$ is the false positive rate based on LogitGap with threshold $\lambda_{\text{L}}$. Similarly, $\text{FPR}_{\text{MCM}}(\tau, \lambda_{\text{MCM}})$ is the false positive rate based on MCM with temperature $\tau$ and threshold $\lambda_{\text{MCM}}$.*

*Proof.* Let $Q_x$ denotes the out-of-distribution $P_{x|\text{OOD}}$. By definition, we express the false positive rate of MCM, denoted as $\text{FPR}_{\text{MCM}}(\tau, \lambda_{\text{MCM}})$, in the following:

$$\text{FPR}_{\text{MCM}}(\tau, \lambda_{\text{MCM}}) = Q_x(S_{\text{MCM}}(\boldsymbol{x}; f) > \lambda_{\text{MCM}})$$

$$= Q_x\left(\frac{e^{z_1'/\tau}}{\sum_{j=1}^{k} e^{z_j'/\tau}} > \lambda_{\text{MCM}}\right). \tag{16}$$

---

[4]For ease of derivation, we adopt Equation 12 in place of $S_{\text{LogitGap}}(\boldsymbol{x}; f) = \frac{1}{K-1} \sum_{j=2}^{K} (z_1' - z_j')$, without affecting performance.

Next, we introduce a intermediate OOD score function, LogitGap with softmax (LogitGap_softmax) to bridge the MCM and our LogitGap. We define the score function of LogitGap_softmax as $S_{\text{LogitGap\_softmax}}(\boldsymbol{x}; f) = \frac{\sum_{i=1}^{K}[e^{z'_1/\tau} - e^{z'_i/\tau}]}{K \sum_{j=1}^{K} e^{z'_j/\tau}}$. Therefore, the false positive rate of LogitGap_softmax can be expressed as

$$
\begin{aligned}
\text{FPR}_{\text{LogitGap\_softmax}}(\tau, \lambda_{\text{LM}}) &= Q_x\left(S_{\text{LogitGap\_softmax}}(\boldsymbol{x}; f) > \lambda_{\text{LM}}\right) \\
&= Q_x\left(\frac{\sum_{i=1}^{K}[e^{z'_1/\tau} - e^{z'_i/\tau}]}{K \sum_{j=1}^{K} e^{z'_j/\tau}} > \lambda_{\text{LM}}\right). \\
&= Q_x\left(\frac{e^{z'_1/\tau}}{\sum_{j=1}^{K} e^{z'_j/\tau}} - \frac{1}{K} > \lambda_{\text{LM}}\right) \\
&= Q_x\left(\frac{e^{z'_1/\tau}}{\sum_{j=1}^{K} e^{z'_j/\tau}} > \lambda_{\text{LM}} + \frac{1}{K}\right)
\end{aligned}
\tag{17}
$$

Combining Eq.(16) and Rq.(17), we have $\lambda_{\text{MCM}} = \lambda_{\text{LM}} + \frac{1}{K}$. Similarly, we can express the false positive rate of LogitGap as

$$
\begin{aligned}
\text{FPR}_{\text{LogitGap}}(\lambda_{\text{L}}) &= Q_x\left(S_{\text{LogitGap}}(\boldsymbol{x}; f) > \lambda_{\text{L}}\right) \\
&= Q_x\left(\frac{\sum_{j=1}^{K}(z'_1 - z'_j)}{K} > \lambda_{\text{L}}\right) \\
&= Q_x\left(\frac{\sum_{j=1}^{K}(z'_1 - z'_j)}{\tau * K} > \frac{1}{\tau} * \lambda_{\text{L}}\right)
\end{aligned}
$$

Since the outputs of CLIP are bounded within $[-1, 1]$ [5], it follows that

$$
\sum_{j=1}^{K} (z'_1 - z'_j) \le 2(K - 1).
$$

Furthermore, we have

$$
\begin{aligned}
\text{FPR}_{\text{LogitGap}}(\lambda_{\text{L}}) &= Q_x\left(\frac{\sum_{j=1}^{K}(z'_1 - z'_j)}{\tau * K} > \frac{1}{\tau} * \lambda_{\text{L}}\right) \\
&\le Q_x\left(\frac{2(K-1)}{\tau * K} > \frac{1}{\tau} * \lambda_{\text{L}}\right) \\
&= Q_x\left(\frac{2(K-1)}{\tau} * \frac{1}{K} > \frac{1}{\tau} * \lambda_{\text{L}}\right) \\
&\le Q_x\left(\frac{2(K-1)}{\tau} * \frac{e^{z'_1/\tau}}{\sum_{j=1}^{K} e^{z'_j/\tau}} > \frac{1}{\tau} * \lambda_{\text{L}}\right) \\
&= Q_x\left(\frac{e^{z'_1/\tau}}{\sum_{j=1}^{K} e^{z'_j/\tau}} > \frac{\lambda_{\text{L}}}{2(K-1)}\right)
\end{aligned}
\tag{18}
$$

Next, for $\lambda_{\text{LM}}$ and $\lambda_{\text{L}}$, we consider two cases: (1) $\lambda_{\text{LM}} \le \frac{\lambda_{\text{L}}}{\tau}$; (2) $\lambda_{\text{LM}} > \frac{\lambda_{\text{L}}}{\tau}$.

For the case (1), with the assumption $\lambda_{\text{LM}} \le \frac{\lambda_{\text{L}}}{\tau}$, we have

$$
\text{FPR}_{\text{LogitGap}}(\lambda_{\text{L}}) \le Q_x\left(\frac{e^{z'_1/\tau}}{\sum_{j=1}^{K} e^{z'_j/\tau}} > \frac{\tau * \lambda_{\text{LM}}}{2(K-1)}\right)
\tag{19}
$$

---

[5] For non-CLIP models, assuming the output range is $[-A, A]$, we similarly obtain $\sum_{j=1}^{K}(z'_1 - z'_j) \le 2A(K-1)$

By comparing Eq.(16) and Eq.(19), we can derive that $\mathrm{FPR}_{\mathrm{LogitGap}}(\lambda_{\mathrm{L}}) \leq \mathrm{FPR}_{\mathrm{MCM}}(\tau, \lambda_{\mathrm{MCM}})$, if the following condition holds:

$$\frac{\tau * \lambda_{\mathrm{LM}}}{2(K-1)} \geq \lambda_{\mathrm{MCM}}. \tag{20}$$

Note that, $\lambda_{LM} = \lambda_{MCM} - \frac{1}{K}$. Combining this equation with Eq.(20), we have

$$\tau \geq 2(K-1) * \frac{\lambda_{\mathrm{MCM}}}{\lambda_{\mathrm{LM}}} > 2(K-1) \tag{21}$$

For the case (2), by directly comparing Eq.(16) and Eq.(18), we can derive that $\mathrm{FPR}_{\mathrm{LogitGap}}(\lambda_{\mathrm{L}}) \leq \mathrm{FPR}_{\mathrm{MCM}}(\tau, \lambda_{\mathrm{MCM}})$, if the following condition holds:

$$\frac{\lambda_{\mathrm{L}}}{2(K-1)} \geq \lambda_{\mathrm{MCM}}. \tag{22}$$

In the case (2), we have $\lambda_{\mathrm{LM}} > \frac{\lambda_{\mathrm{L}}}{\tau}$. With this inequation, Eq.(22) can be rewitten as

$$\frac{\tau * \lambda_{\mathrm{LM}}}{2(K-1)} > \frac{\lambda_{\mathrm{L}}}{2(K-1)} \geq \lambda_{\mathrm{MCM}}. \tag{23}$$

Note that Eq.(23) is the same as Eq.(21), thus we can derive the same conclusion. $\qquad\square$

# B  Analysis: Logits Distribution Differences Between ID and OOD Samples

## B.1  Motivation Example

To illustrate a key limitation of the softmax-based scoring function MCM [33], we present two examples where *logit distributions differ significantly* yet yield *nearly identical MCM scores*. This reveals how the softmax operation can suppress informative structural differences in the logit space, which are often critical for distinguishing ID and OOD samples. Consider a 3-way classification problem with two test samples, $x^1$ and $x^2$, whose corresponding logit vectors are $z^1 = [0.5596, -0.9808, -0.9808]$ and $z^2 = [0.9783, -0.6311, -0.4976]$, respectively. Despite the significant difference in magnitude and sharpness ($z^2$ exhibits a much more confident prediction), the maximum softmax probability for both is identical: $\mathrm{softmax}\left(z^1\right) = [\mathbf{0.70}, 0.15, 0.15]$ and $\mathrm{softmax}\left(z^2\right) = [\mathbf{0.70}, 0.14, 0.16]$.

This arises because softmax normalizes logits into a probability distribution, preserving their relative ordering while discarding absolute magnitudes. As a result, distinct logit patterns can be mapped to probability vectors with similar maximum values, thereby limiting the discriminative power of softmax-based OOD detection. This issue is further exacerbated by temperature scaling $\tau > 1$, which flattens the softmax distribution and amplifies information loss. This observation naturally raise a key question: *How can we more effectively leverage the discriminative information embedded in non-maximum logits to enhance ID-OOD separability?*

## B.2  Motivation Evidence

We observe an interesting phenomenon: the relationship between the maximum logit and the remaining logits differs notably between in-distribution (ID) and out-of-distribution (OOD) samples. As shown in Figure 5, ID samples tend to have a larger maximum logit value than OOD samples. Conversely, for the non-maximum logits, OOD samples typically exhibit higher values than ID samples.

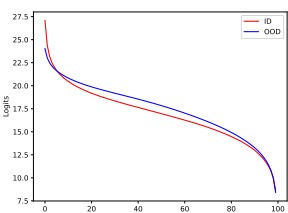

To verify the generality of this phenomenon, we conduct experiments across various ID/OOD datasets and model architectures in Table 7. Specifically, we report the average maximum logit, and the average logit of non-predicted classes, using ResNet-50, ViT-B/16, and ViT-L/14 as backbone models.

Figure 5: Descending-sorted logits on CLIP ViT-B/16 with ImageNet100 (ID) and iNaturalist (OOD).

## B.3  Theoretical Analysis

Our theoretical motivation stems from the observed distinction in logit patterns between ID and OOD samples: (i) Higher maximum logit values for ID samples; (ii) Higher non-predicted class logits in OOD samples. This indicates that the gap between the maximum logit and the remaining logits is generally wider for ID samples than for OOD samples.

To understand the rationale behind this, we provide a simple analysis from the theoretical perspective.

**Theorem B.1.** *In a binary classification problem, given a well-trained feature extractor $\phi$ and a classifier $W$, we assume that the feature distribution of ID samples for class $i$ follows a Gaussian distribution, $\phi(x_{\mathbf{ID}}|y = 0) \sim \mathcal{N}(\mu_0, \Sigma_0)$ and $\phi(x_{\mathbf{ID}}|y = 1) \sim \mathcal{N}(\mu_1, \Sigma_1)$. We further assume that the features of OOD samples can be modeled as an interpolation of two ID feature distribution with a Gaussian noise, i.e.$\phi(x_{\mathbf{OOD}}) = \alpha \cdot \mathcal{N}(\mu_1, \Sigma_1) + (1 - \alpha) \cdot \mathcal{N}(\mu_0, \Sigma_0) + \beta \cdot \mathcal{N}(0, \Sigma)$. Let $z^{\mathbf{ID}} = [z_0^{\mathrm{ID}}, z_1^{\mathrm{ID}}]$ denotes the predicted logit vector for ID sample $x_{\mathbf{ID}}$, $z^{\mathbf{OOD}} = [z_0^{\mathrm{OOD}}, z_1^{\mathrm{OOD}}]$ denotes the predicted logit vector for OOD sample $x_{\mathbf{OOD}}$. If the ID sample $x_{\mathbf{ID}}$ belongs to class $i$, we have*

$$\mathbb{E}[z_{1-i}^{\mathrm{ID}}] < \mathbb{E}[z_{1-i}^{\mathrm{OOD}}] \tag{24}$$

*Proof.* If the ID sample $x_{\mathbf{ID}}$ belongs to class $i$, we have

$$\mathbb{E}[z_{1-i}^{\mathrm{ID}}] = w_{1-i}^T \mathbb{E}[\phi(x_{\mathbf{ID}})] = w_{1-i}^T \mu_i \tag{25}$$

Table 7: Logit distribution differences between ID and OOD data. $z'_1$ and $\bar{z}'_N$ denote the maximum logit and the average logit across all non-predicted classes, respectively.

|  | $z'_1$ | $\bar{z}'_N$ |
|---|---|---|
| CIFAR-10 (ID) | 9.02 | -0.12 |
| CIFAR-100 | 7.69 | 0.40 |
| TIN | 7.19 | 0.57 |
| Textures | 5.54 | 0.85 |
| SVHN | 5.51 | 0.84 |

(a) Logit distribution on ResNet-50 using CIFAR-10 as ID dataset.

|  | $z'_1$ | $\bar{z}'_N$ |
|---|---|---|
| ImageNet (ID) | 30.82 | 17.32 |
| iNaruralist | 26.48 | 17.89 |
| SUN | 26.71 | 17.90 |
| Textures | 28.72 | 18.94 |

(b) Logit distribution on CLIP ViT-B/16 using ImageNet as ID dataset.

|  | $z'_1$ | $\bar{z}'_N$ |
|---|---|---|
| ImageNet (ID) | 25.76 | 11.19 |
| iNaturalist | 20.72 | 11.65 |
| Textures | 23.85 | 13.18 |
| ImageNet-O | 21.14 | 12.16 |

(c) Logit distribution on CLIP ViT-L/14 using ImageNet as ID dataset.

Similarly, we have

$$\mathbb{E}[z_{1-i}^{\text{OOD}}] = w_{1-i}^T \mathbb{E}[\phi(\boldsymbol{x_{\text{OOD}}})]. \tag{26}$$

Assuming the features of $\boldsymbol{x_{\text{OOD}}}$ can be denoted as the interpolation of two ID feature distribution with a random Gaussian noise, $i.e. \phi(\boldsymbol{x_{\text{OOD}}}) = \alpha \cdot \mathcal{N}(\mu_1, \Sigma_1) + (1-\alpha) \cdot \mathcal{N}(\mu_0, \Sigma_0) + \beta \cdot \mathcal{N}(0, \Sigma)$, we have

$$\begin{aligned}
\mathbb{E}[z_{1-i}^{\text{OOD}}] &= w_{1-i}^T \mathbb{E}[\phi(\boldsymbol{x_{\text{OOD}}})] \\
&= w_{1-i}^T (\alpha \cdot \mu_1 + (1-\alpha) \cdot \mu_0) \\
&= \alpha \cdot w_{1-i}^T \mu_1 + (1-\alpha) \cdot w_{1-i}^T \mu_0.
\end{aligned} \tag{27}$$

We assume that the classifier $W$ is well trained, thus $w_{1-i}^T \mu_i < w_{1-i}^T \mu_{1-i}$. Then, we have

$$w_{1-i}^T \mu_i < \alpha \cdot w_{1-i}^T \mu_1 + (1-\alpha) w_{1-i}^T \mu_0. \tag{28}$$

Therefore, we have

$$\mathbb{E}[z_{1-i}^{\text{ID}}] < \mathbb{E}[z_{1-i}^{\text{OOD}}]. \tag{29}$$

$\square$

# C   Experimental Details

## C.1   Implementation Details

We run all OOD detection experiments on NVIDIA GeForce RTX-4090Ti GPUs with Pytorch 2.3.1. For CLIP-based OOD detection, we adopt the pre-trained ViT-B/16 [10] model from CLIP [39]. For conventional OOD detection task using CIFAR-10 [24] as ID dataset, we use a ResNet-50 [13] model pre-trained on the CIFAR-10 training set as the backbone.

## C.2   Datasets

**ImageNet-10, ImageNet-20, ImageNet100**   [33] creates ImageNet-10 that mimics the class distribution of CIFAR-10 [24] but with high-resolution images. For semantically hard OOD evaluation with realistic datasets, [33] curates ImageNet-20, which consists of 20 classes semantically similar to ImageNet-10 (e.g., dog (ID) vs. wolf (OOD)). ImageNet-100 is created by randomly sample 100 classes from ImageNet-1k [7].

**NINCO**   The NINCO [2] main dataset comprises 64 OOD classes with a total of 5,879 samples. These classes were carefully selected to ensure no categorical overlap with any of the ImageNet-1K [7] classes. Additionally, each sample was manually inspected by the authors to confirm the absence of ID objects, making NINCO a reliable benchmark for evaluating out-of-distribution detection on ImageNet-1K.

**ImageNetOOD**   ImageNet-OOD [55] is a clean, manually curated, and diverse dataset containing 31,807 images from 637 classes, designed for evaluating semantic shift detection using ImageNet-1K [7] as the in-distribution (ID) dataset. To minimize covariate shifts, images are sourced directly from ImageNet-21K [41], with human verification ensuring the removal of any ID contamination from ImageNet-1K. The dataset also addresses multiple sources of semantic ambiguity caused by inaccurate hierarchical relationships in ImageNet labels and eliminates visually ambiguous images stemming from inconsistencies in ImageNet's data curation process.

**ImageNet-O**   ImageNet-O is a dataset containing image concepts absent from ImageNet-1K [7], specifically designed to evaluate the robustness of ImageNet models. These out-of-distribution (OOD) images consistently cause models to misclassify them as high-confidence in-distribution examples. As the first anomaly and OOD dataset tailored for testing ImageNet-1K models, ImageNet-O provides a valuable benchmark for assessing OOD detection performance under label distribution shifts.

**ImageNet-A**   ImageNet-A [19] contains images sampled from a distribution distinct from the standard ImageNet-1k [7] training distribution. Although its examples belong to existing ImageNet-1k classes, they are deliberately more challenging, frequently leading to misclassifications across a range of models. ImageNet-A enables we to test image classification performance when the input data has covariate distribution shifts.

**ImageNet-R**   ImageNet-R (Rendition) [16] consists of artistic and non-photographic renditions of ImageNet-1k [7] classes, including art, cartoons, graffiti, embroidery, graphics, origami, paintings, patterns, plastic figures, plush toys, sculptures, sketches, tattoos, video game assets, and more. The dataset covers 200 ImageNet-1k classes with a total of 30,000 images, offering a diverse benchmark for evaluating model robustness to distribution shifts in visual style and appearance.

**ImageNet-Sketch**   The ImageNet-Sketch [49] dataset contains 50,889 images, with approximately 50 images per class for all 1,000 ImageNet-1k [7] categories. It was constructed by querying Google Images using the phrase "sketch of [class name]," restricted to a black-and-white color scheme. An initial set of 100 images was collected for each class, followed by manual filtering to remove irrelevant images and those belonging to visually similar but incorrect classes. For categories with fewer than 50 valid images after cleaning, data augmentation techniques such as flipping and rotation were applied to balance the dataset.

Table 8: The value of $N$ for two methods on different datasets.

| | $K$ | $N$ (Fixed) | $N$ (Adaptive) |
|---|---|---|---|
| ImageNet | 1000 | 200 | 88 |
| ImageNet-100 | 100 | 20 | 20 |
| ImageNet-10 | 10 | 5 | 4 |
| ImageNet-20 | 20 | 10 | 6 |

## C.3 Logits Selection Methods

**Logits Selection with a Fixed Hyperparameter $N$**    The effectiveness of our proposed LogitGap method relies on the selection of an informative region within the logits space, which is governed by a hyperparameter $N$. To determine an appropriate value for $N$, we consider the number of classes in the dataset. For datasets with a large number of classes (e.g., ImageNet and ImageNet-100), we set $N$ to 20% of the total number of classes $K$. In contrast, for datasets with fewer classes (e.g., ImageNet-10 and ImageNet-20), we set $N$ to 50% of $K$. This strategy is motivated by the following intuition: when $K$ is large, using a lower proportion (e.g., 20%) helps reduce the influence of noise; whereas when $K$ is small, a higher proportion (e.g., 50%) preserves more useful information. Moreover, our empirical results demonstrate a consistent performance improvement when $N$ is set within the 20%–50% range of the total number of classes across different datasets. The specific $N$ values chosen for different datasets are summarized in Table 8.

**Logits Selection with an Adaptive Hyperparameter $N$**    To enhance LogitGap, we introduce an improved strategy to adaptively choice the value of $N$. To this end, we firstly construct a small validation set randomly sampled from the in-distribution (ID) data, with a fixed size of 100 samples. Using the model's feature extractor, we obtain the image features of these ID samples. We then perform random inter-class interpolation on these features to generate synthetic OOD samples. To further increase the diversity of these synthetic samples, Gaussian noise is added to the interpolated features. The interpolation process is formally defined as follows:

$$x_{\text{OOD}} = \alpha \cdot x_i + (1 - \alpha) \cdot x_j + \beta \cdot \mathcal{N}(0, I), \quad (30)$$

where $x_i$ and $x_j$ represent two samples from the ID validation set, $\alpha$ denotes the mixing coefficient between samples, while $\beta$ controls the weight of the added noise.

In our experiments, we generate a set of synthetic OOD samples equal in size to the ID validation set to serve as the OOD validation set. For datasets with a large number of categories, such as ImageNet [7] and ImageNet-100 [33], we set the interpolation parameters to $\alpha = 0.3$ and $\beta = 0.8$. For smaller-scale datasets like ImageNet-10 [33] and ImageNet-20 [33], we set $\alpha = 0.3$ and $\beta = 0.0$. We then compute the logits scores for both the ID and synthetic OOD validation samples and determine the optimal value of $N$ adaptively based on the following criterion:

$$\arg \max_N \left( \mathbb{E}_{x \sim P_{\text{OOD}}}[\bar{z}'_N] - \mathbb{E}_{x \sim P_{\text{ID}}}[\bar{z}'_N] \right), \quad (31)$$

where $\bar{z}'_N$ represents the mean of the logits ranked from second to the $N$-th largest.

## C.4 Implementation Details on LogitGap Combined with Negative-Prompt-Based OOD Method

Negative-Prompt-Based OOD detection methods jointly train both positive and negative prompts to separately capture the characteristics of ID and OOD samples. In these methods, the model's predicted logits take the form $z = [z_1^{\text{ID}}, z_2^{\text{ID}}, ..., z_K^{\text{ID}}, z_1^{\text{OOD}}, z_2^{\text{OOD}}, ..., z_M^{\text{OOD}}]$, where $z_i^{\text{ID}}$ denotes the logit for the $i$-th ID class, and $z_j^{\text{OOD}}$ denotes the logit for the $j$-th OOD class. Since negative prompts tend to increase the logits of OOD samples, directly applying LogitGap over the entire logits space is not appropriate. In our experiments, when combining ID-Like [1] with LogitGap, we rely solely on the ID logits, $z^{\text{ID}} = [z_1^{\text{ID}}, z_2^{\text{ID}}, ..., z_K^{\text{ID}}]$, to compute the LogitGap score.

## D Experimental Results

### D.1 Results on Traditional OOD Detection with CIFAR100 as ID Dataset

To further evaluate the effectiveness and generality of our LogitGap, we conduct experiments in traditional OOD detection settings with CIFAR100 [24] as ID dataset. Following the OpenOOD [54] protocol, we adopt the standard OpenOOD benchmark splits for OOD evaluation. The OOD benchmarks include both **near-OOD datasets**: CIFAR-10 [24], Tiny ImageNet [46], and **far-OOD datasets**: MNIST [8], SVHN [36], Textures [5], and Places365 [61]. In this setup, we employ a ResNet-50 [13] backbone trained from scratch on ID data using cross-entropy loss. We compare LogitGap against a comprehensive suite of baselines: (i) Logit-based post-hoc methods: MSP [17], MaxLogit [14], KL-Matching [15], ODIN [27], IODIN [40]; (ii) Internal network statistics-based methods: Mahananobis [25], ASH [9], SHE (Hopfield energy) [58]; (iii) Training-based methods with auxiliary data: Outlier Exposure (OE) [18], MixOE [57]. The results are summarized in Table 9.

As shown in Table 9, our proposed LogitGap demonstrates superior performance across all evaluated benchmarks. We observe that: (1) Among logit-based methods, IODIN [40] achieves the best performance. ODIN [27] enhances OOD detection by introducing input perturbations to amplify the difference between ID and OOD samples in the softmax output. Building upon this, IODIN proposes to mask low-magnitude regions and perturb only invariant features, effectively encouraging the model to ignore environmental factors and focus on more informative regions. This requires calculating an additional invariant feature mask on top of the gradient computation in ODIN. In contrast, LogitGap improves detection performance by applying a simple transformation in the output space, reducing the FPR95 by 1.98% and 0.78% compared to MSP [17] and IODIN, respectively. (2) Internal network statistics-based methods generally perform worse because they heavily rely on the representation quality and discriminative ability of the internal network. When the backbone lacks sufficient expressive power, the extracted feature distributions of ID and OOD samples tend to overlap, making it difficult to effectively distinguish between them. (3) When combined with methods that leverage extra out-of-distribution data, such as OE [18] and MixOE [57], LogitGap further enhances performance over their respective baselines. For example, a straightforward substitution of the OOD score function with LogitGap leads to a 3.57% and 1.23% performance gain in FPR95 and AUROC respectively over the original OE [18] method. These experiments highlight the advantages of LogitGap in terms of ease of integration and wide applicability across different settings.

Table 9: OOD Detection performance on ResNet-50 using CIFAR-100 as the ID dataset.

| Method | Near OOD | | | | Far OOD | | | | | | | | AVG | |
| --- | --- | --- | --- | --- | --- | --- | --- | --- | --- | --- | --- | --- | --- | --- |
| | CIFAR10 | | TIN | | MNIST | | SVHN | | Textures | | PLACES365 | | | |
| | FPR95 | AUROC | FPR95 | AUROC | FPR95 | AUROC | FPR95 | AUROC | FPR95 | AUROC | FPR95 | AUROC | FPR95 ↓ | AUROC ↑ |
| MSP [17] | 58.90 | 78.47 | 50.70 | 82.07 | 57.23 | 76.08 | 59.07 | 78.42 | 61.89 | 77.32 | 56.63 | 79.23 | 57.40 | 78.60 |
| MaxLogit [14] | 59.11 | 79.21 | 51.83 | 82.90 | 52.94 | 78.91 | 53.90 | 81.65 | 62.39 | 78.39 | 57.67 | 79.75 | 56.31 | 80.14 |
| ODIN [27] | 60.63 | 78.18 | 55.21 | 81.63 | **45.94** | **83.79** | 67.43 | 74.54 | 62.37 | 79.34 | 59.73 | 79.45 | 58.55 | 79.49 |
| IODIN [40] | 59.09 | 79.24 | 51.57 | 82.96 | 52.93 | 78.89 | 54.06 | 81.56 | 62.07 | 78.48 | 57.47 | 79.83 | 56.20 | 80.16 |
| KL-Matching [15] | 84.77 | 73.92 | 70.99 | 79.21 | 72.88 | 74.15 | 50.31 | 79.32 | 81.80 | 75.76 | 81.62 | 75.68 | 73.73 | 76.34 |
| Mahalanobis [25] | 88.00 | 55.87 | 79.04 | 61.50 | 71.71 | 67.47 | 67.22 | 70.67 | 70.49 | 76.26 | 79.60 | 63.15 | 76.01 | 65.82 |
| ASH [9] | 68.07 | 76.47 | 63.37 | 79.92 | 66.58 | 77.23 | **45.98** | **85.60** | 61.29 | **80.72** | 62.96 | 78.76 | 61.38 | 79.78 |
| SHE [58] | 60.41 | 78.15 | 57.73 | 79.74 | 58.78 | 76.76 | 59.16 | 80.97 | 73.28 | 73.64 | 65.26 | 76.30 | 62.44 | 77.59 |
| LogitGap | 58.70 | 79.43 | 50.01 | 83.31 | 52.49 | 78.60 | 54.84 | 81.07 | 60.34 | 78.86 | 56.12 | 80.41 | 55.42 | 80.28 |
| OE [18] | 63.87 | 74.64 | **0.41** | **99.88** | 35.03 | 91.28 | 56.24 | 85.73 | 52.83 | 83.98 | 60.49 | 76.89 | 44.81 | 85.40 |
| +LogitGap | 63.29 | 75.73 | 0.42 | 99.88 | 24.96 | 92.93 | 50.28 | 87.68 | 49.82 | 85.37 | 58.67 | 78.21 | 41.24 | 86.63 |
| MixOE [57] | **61.09** | 78.18 | 49.43 | 83.93 | 68.04 | **70.06** | 76.72 | 73.06 | 66.37 | 78.03 | 56.10 | 80.44 | 62.96 | 77.28 |
| +LogitGap | 61.31 | **78.41** | 48.41 | 84.05 | 67.43 | 69.98 | 75.10 | 73.22 | 65.58 | 78.22 | 55.31 | 80.69 | 62.19 | 77.43 |

### D.2 Results on Traditional OOD Detection with ImageNet as ID Dataset

In the traditional OOD detection setting, we further evaluate the performance of LogitGap using a ResNet-50 [13] backbone on the large-scale ImageNet [7] dataset. Following the OpenOOD [54] protocol, we adopt SSB-hard [48] and NINCO [2] as **near-OOD** datasets, iNaturalist[47], Textures[5], and OpenImage-O [50] as **far-OOD** datasets. In this setup, the ResNet-50 model [13] is trained from scratch on the ID data using cross-entropy loss. We compare LogitGap against several representative

Table 10: OOD Detection performance on ResNet-50 using ImageNet as the ID dataset.

| Method | Near OOD | | | | Far OOD | | | | | | AVG | |
| | SSB-hard | | NINCO | | iNaturalist | | Textures | | OpenImage-O | | | |
| | FPR95 | AUROC | FPR95 | AUROC | FPR95 | AUROC | FPR95 | AUROC | FPR95 | AUROC | FPR95 | AUROC |
|---|---|---|---|---|---|---|---|---|---|---|---|---|
| MSP [17] | **73.6** | **73.2** | 54.13 | 82.05 | 29.55 | 91.70 | 50.59 | 87.21 | 37.44 | 88.94 | 49.06 | 84.62 |
| MaxLogit [14] | 76.19 | 72.51 | 59.49 | 80.41 | 30.59 | 91.16 | 46.12 | 88.39 | 37.88 | **89.17** | 50.05 | 84.33 |
| KL-Matching [15] | 84.72 | 71.38 | 60.28 | 81.90 | 38.51 | 90.79 | 52.38 | 84.72 | 48.94 | 87.30 | 56.97 | 83.22 |
| ODIN [27] | 76.86 | 71.74 | 68.08 | 77.77 | 36.12 | 91.16 | 49.25 | 89.01 | 46.48 | 88.23 | 55.36 | 83.58 |
| GradNorm [21] | 78.24 | 71.90 | 79.50 | 74.02 | 32.01 | **93.89** | **43.24** | **92.05** | 68.46 | 84.83 | 60.29 | 83.34 |
| Energy [30] | 76.53 | 72.08 | 60.61 | 79.70 | 31.33 | 90.63 | 45.77 | 88.70 | 38.07 | 89.06 | 50.46 | 84.03 |
| LogitGap | 75.48 | 72.51 | **53.64** | **82.25** | **26.68** | 92.24 | 47.11 | 87.15 | **35.10** | 89.11 | **47.60** | **84.65** |

Table 11: OOD Detection performance on CLIP ViT-B/16 under zero-shot setting. Results are reported with ImageNet as the ID dataset in the far-OOD scenario.

| Method | OOD Dataset | | | | | | | | | AVG | | |
| | iNaturalist | | | Textures | | | OpenImage-O | | | | | |
| | FPR95 ↓ | AUROC ↑ | AUPR ↑ | FPR95 ↓ | AUROC ↑ | AUPR ↑ | FPR95 ↓ | AUROC ↑ | AUPR ↑ | FPR95 ↓ | AUROC ↑ | AUPR ↑ |
|---|---|---|---|---|---|---|---|---|---|---|---|---|
| | | | | | | Zero-shot | | | | | | |
| Energy [30] | 73.86 | 87.08 | 97.28 | 92.71 | 66.08 | 94.77 | 65.60 | 85.91 | 94.55 | 77.39 | 79.69 | 95.53 |
| MCM [33] | 31.49 | 94.39 | 98.80 | 58.76 | **85.84** | **98.02** | 40.76 | 91.99 | 96.95 | 43.67 | 90.74 | 97.92 |
| MaxLogit [14] | 56.25 | 90.47 | 97.99 | 86.45 | 71.85 | 95.66 | 51.89 | 88.93 | 95.66 | 64.86 | 83.75 | 96.44 |
| LogitGap | 32.54 | 94.13 | 98.74 | 58.56 | 85.73 | 97.96 | 39.12 | 92.41 | 97.11 | 43.41 | 90.76 | 97.94 |
| LogitGap* | **27.82** | **94.89** | **98.91** | **58.17** | 85.68 | 97.97 | **37.27** | **92.66** | **97.19** | **41.09** | **91.08** | **98.02** |

post-hoc OOD detection methods, including MSP [17], MaxLogit [14], KL-Matching [15], ODIN [27], IODIN [40], Energy [30], and GradNorm [21].

As summarized in Table 10, LogitGap achieves comparable or superior performance across all benchmarks and obtains the best overall average performance, demonstrating its robustness and effectiveness in the traditional OOD detection scenario.

## D.3 Results on Far-OOD Evaluation with ImageNet as ID Dataset in Zero-Shot OOD Detection

In the main paper, to isolate the effect of semantic shift, we construct a challenging OOD detection benchmark specifically designed for semantic shift evaluation. This benchmark is built using ImageNet [7] as ID dataset, while NINCO [2], ImageNet-O [19], and ImageNet-OOD [55] as OOD datasets, enabling a more accurate assessment of the model's ability to detect semantic-level distribution changes.

Moreover, following the OpenOOD [54] evaluation protocol, we assess the performance of OOD detection methods in the far-OOD scenario under a zero-shot setting. In the far-OOD scenario, ImageNet is used as the ID dataset, while iNaturalist [47], OpenImage-O [50], and Textures [5] serve as the OOD datasets. As shown in Table 11, LogitGap maintains robust performance under the broader distributional shift.

## D.4 Results on More Architectures with ImageNet as ID Dataset in Zero-Shot OOD Detection

We extend our experiments to other CLIP variants with ImageNet as ID dataset, including ResNet-50, ResNet-101, ViT-B/32 and ViT-L/14 in zero-shot OOD detection. As shown in Table 12, LogitGap consistently outperforms baselines across all backbones. For instance, on ViT-L/14, LogitGap reduces FPR95 by 6.03% and improves AUROC by 2.65% compared to MCM. This demonstrate that LogitGap generalizes well across architectures.

Table 12: OOD Detection performance across various CLIP architectures under the zero-shot setting. Results are reported with ImageNet as the ID dataset in a semantic shift scenario.

| | NINCO | | | OOD Dataset ImageNet-O | | | ImageNetOOD | | | AVG | | |
|---|---|---|---|---|---|---|---|---|---|---|---|---|
| Method | FPR95 ↓ | AUROC ↑ | AUPR ↑ | FPR95 ↓ | AUROC ↑ | AUPR ↑ | FPR95 ↓ | AUROC ↑ | AUPR ↑ | FPR95 ↓ | AUROC ↑ | AUPR ↑ |
| Energy [30] | 92.21 | 66.35 | 94.67 | 86.00 | 70.98 | 98.32 | 80.87 | 74.85 | 82.57 | 86.36 | 70.73 | 91.85 |
| MCM [33] | **82.09** | 71.58 | 95.55 | 84.25 | 73.95 | 98.57 | 84.97 | 75.19 | 83.64 | 83.77 | 73.57 | 92.59 |
| MaxLogit [14] | 88.19 | 69.42 | 95.23 | 83.95 | 72.50 | 98.41 | **78.74** | **76.64** | 83.91 | 83.63 | 72.85 | 92.52 |
| LogitGap | 82.65 | **73.26** | **95.87** | **83.70** | **75.44** | **98.67** | 84.01 | 76.51 | **84.35** | **83.45** | **75.07** | **92.96** |

(a) Results on ResNet-50.

| | NINCO | | | OOD Dataset ImageNet-O | | | ImageNetOOD | | | AVG | | |
|---|---|---|---|---|---|---|---|---|---|---|---|---|
| Method | FPR95 ↓ | AUROC ↑ | AUPR ↑ | FPR95 ↓ | AUROC ↑ | AUPR ↑ | FPR95 ↓ | AUROC ↑ | AUPR ↑ | FPR95 ↓ | AUROC ↑ | AUPR ↑ |
| Energy [30] | 91.83 | 66.06 | 94.62 | 86.15 | 70.51 | 98.32 | 81.19 | 75.07 | 83.21 | 86.39 | 70.55 | 92.05 |
| MCM [33] | 84.13 | 70.66 | 95.38 | 83.75 | 74.06 | 98.56 | 85.19 | 74.76 | 83.11 | 84.36 | 73.16 | 92.35 |
| MaxLogit [14] | 87.94 | 69.41 | 95.24 | 84.15 | 72.46 | 98.43 | 79.73 | **76.83** | **84.37** | 83.94 | 72.90 | 92.68 |
| LogitGap | **82.65** | **73.36** | **95.89** | **82.10** | **75.94** | **98.69** | 83.99 | 76.18 | 83.88 | **82.91** | **75.16** | **92.82** |

(b) Results on ResNet-101.

| | NINCO | | | OOD Dataset ImageNet-O | | | ImageNetOOD | | | AVG | | |
|---|---|---|---|---|---|---|---|---|---|---|---|---|
| Method | FPR95 ↓ | AUROC ↑ | AUPR ↑ | FPR95 ↓ | AUROC ↑ | AUPR ↑ | FPR95 ↓ | AUROC ↑ | AUPR ↑ | FPR95 ↓ | AUROC ↑ | AUPR ↑ |
| Energy [30] | 87.05 | 68.75 | 94.96 | 83.80 | 72.46 | 98.39 | 79.66 | 75.12 | 82.67 | 83.50 | 72.11 | 92.01 |
| MCM [33] | 79.62 | 73.85 | 95.93 | 80.95 | 75.58 | 98.66 | 84.09 | 76.32 | 84.16 | 81.55 | 75.25 | 92.92 |
| MaxLogit [14] | 82.27 | 71.76 | 95.53 | 80.55 | 74.32 | 98.51 | **77.38** | 76.99 | 84.06 | **80.07** | 74.36 | 92.70 |
| LogitGap | **79.13** | **75.66** | **96.27** | **79.15** | **77.45** | **98.77** | 82.03 | **77.68** | **84.84** | 80.10 | **76.93** | **93.29** |

(c) Results on ViT-B/32.

| | NINCO | | | OOD Dataset ImageNet-O | | | ImageNetOOD | | | AVG | | |
|---|---|---|---|---|---|---|---|---|---|---|---|---|
| Method | FPR95 ↓ | AUROC ↑ | AUPR ↑ | FPR95 ↓ | AUROC ↑ | AUPR ↑ | FPR95 ↓ | AUROC ↑ | AUPR ↑ | FPR95 ↓ | AUROC ↑ | AUPR ↑ |
| Energy [30] | 81.88 | 76.89 | 96.51 | 77.45 | 78.45 | 98.81 | 76.95 | 78.16 | 84.61 | 78.76 | 77.83 | 93.31 |
| MCM [33] | 69.48 | 79.13 | 96.74 | 68.30 | 82.48 | 99.06 | 73.99 | 80.71 | 86.60 | 70.59 | 80.77 | 94.13 |
| MaxLogit [14] | 74.23 | 79.18 | 96.86 | 71.85 | 80.73 | 98.94 | 72.78 | 79.95 | 85.69 | 72.95 | 79.95 | 93.83 |
| LogitGap | **64.43** | **82.49** | **97.38** | **62.05** | **84.94** | **99.20** | **67.20** | **82.84** | **87.75** | **64.56** | **83.42** | **94.78** |

(d) Results on ViT-L/14.

# E More Analysis about LogitGap

## E.1 Generalization Beyond Visual Tasks

Although LogitGap is primarily designed for visual classification, one of the core settings for OOD detection, we further verify its generalization to non-visual tasks. Specifically, we conduct an evaluation on the ESC-50 [38] audio classification dataset, where 50 categories are randomly divided into 25 ID and 25 OOD classes. The method is implemented with the CLAP [12] model, following the same hyperparameter selection strategy used in the visual domain, with $N = 20\%$ of the class count ($N = 5$).

As shown in Table 13, LogitGap maintains strong performance on this audio task, indicating that it does not rely on modality-specific architectures or features. This confirms its broad applicability across domains. In addition, we introduce MCM_topN, a comparative baseline that applies the top-N strategy to MCM [33]. For fair comparison, we use the same $N$ for MCM_topN and LogitGap. Results show that applying the top-N strategy to MCM does not consistently improve performance, which indicates LogitGap's advantage goes beyond top-N filtering.

Table 13: OOD Detection performance on CLAP using ESC-50 dataset, where 50 classes are randomly split into 25 ID and 25 OOD classes.

|  | FPR95 ↓ | AUROC ↑ | AUPR ↑ |
|---|---|---|---|
| MCM [33] | 26.40 | 94.86 | 95.35 |
| MaxLogit [14] | 40.60 | 92.32 | 92.51 |
| Energy [30] | 42.10 | 91.76 | 92.09 |
| MCM_topN | 27.90 | 94.69 | 95.21 |
| GEN [32] | 24.10 | 95.55 | 95.96 |
| LogitGap | **17.50** | **96.15** | **96.48** |

## E.2 Relationship Between LogitGap and Other Logit-Pattern-Based Methods

We review related works in both active learning and OOD detection, and summarize several representative methods in Table 14. Here, $z_c$ denotes the logit and $p_c$ represents the predicted probability for $c$-th class. Specifically, we compare LogitGap with three classical active learning approaches: LC [6], Margin [42], and Entropy [43]; and four representative OOD detection methods: MSP [17], MaxLogit[14], DML [60], and GEN [32].

Among these methods, Margin Sampling is the most closely related to ours, as it also measures a margin between model outputs. However, two key distinctions set LogitGap apart: (1) Representation Level: Margin Sampling operates on predicted probabilities, whereas LogitGap computes directly on raw logits, avoiding softmax-induced normalization effects. (2) Scope of Comparison: Margin Sampling considers only the top-2 predictions, while LogitGap generalizes this to the top-N logits, enabling a more holistic uncertainty estimation.

Table 14: Comparison of logit-pattern-based methods, "OOD" and "AL" represent OOD detection and active learning respectively.

| Method | Task | Equation |
|---|---|---|
| MSP [17] | OOD | $\max_c p_c$ |
| MaxLogit [14] | OOD | $\max_c z_c$ |
| DML [60] | OOD | $\max_c \lambda \hat{z}_c + \|z_c\|, z_c = \hat{z}_c \cdot \|z_c\|$ |
| GEN [32] | OOD | $-\sum_{c=1}^{M} p_c^\gamma (1 - p_c)^\gamma$, and $p_1 \geq \cdots \geq p_M \geq \cdots \geq p_N, \gamma \in (0, 1)$ |
| Margin [42] | AL | $p_1 - p_2$ and $p_1 \geq p_2 ... \geq p_N$ |
| LC [6] | AL | $\max_c 1 - p_c$ |
| Entropy [43] | AL | $-\sum_c p_c \cdot \log p_c$ |
| LogitGap | OOD | $\frac{1}{M-1} \sum_{c=2}^{M} z_1 - z_c$ and $z_1 \geq \cdots \geq z_M \geq \cdots \geq z_N$ |

Table 15: OOD Detection Performance on CLIP ViT-B/16 under Zero-shot Setting. Results are reported with ImageNet as the ID dataset in a semantic shift scenario.

| | NINCO | | | OOD Dataset ImageNet-O | | | ImageNetOOD | | | AVG | | |
|---|---|---|---|---|---|---|---|---|---|---|---|---|
| Method | FPR95 ↓ | AUROC ↑ | AUPR ↑ | FPR95 ↓ | AUROC ↑ | AUPR ↑ | FPR95 ↓ | AUROC ↑ | AUPR ↑ | FPR95 ↓ | AUROC ↑ | AUPR ↑ |
| Margin Sampling | 89.88 | 68.40 | 94.97 | 89.45 | 67.83 | 98.14 | 89.96 | 67.49 | 77.67 | 89.76 | 67.91 | 90.26 |
| LogitGap | **77.42** | **76.51** | **96.38** | **71.95** | **81.45** | **99.03** | **75.40** | **80.27** | **86.21** | **74.92** | **79.41** | **93.87** |

Table 16: $N$ selected using synthetic and real OOD samples on ViT-B/16 with ImageNet as ID dataset. $N_{\text{syn}}$ and $N_{\text{real}}$ denote $N$ selected using synthetic OOD and real OOD samples, respectively.

| | NINCO | | | ImageNet-O | | | ImageNetOOD | | |
|---|---|---|---|---|---|---|---|---|---|
| | FPR95 ↓ | AUROC ↑ | $N$ | FPR95 ↓ | AUROC ↑ | $N$ | FPR95 ↓ | AUROC ↑ | $N$ |
| $N_{syn}$ | 77.42 | 76.51 | 88 | 71.95 | 81.45 | 88 | 75.40 | 80.27 | 88 |
| $N_{real}$ | 77.22 | 76.51 | 100 | 71.60 | 81.50 | 110 | 75.39 | 80.27 | 90 |

To emphasize the importance of these distinctions, we adapt Margin Sampling for OOD detection under zero-shot setting using CLIP ViT-B/16, with ImageNet as the ID dataset in a semantic shift scenario. As shown in Table 15, LogitGap consistently outperforms Margin Sampling across all benchmarks, demonstrating the effectiveness and robustness of our formulation.

### E.3 The Effectiveness of Synthetic OOD Data

As described in Section C.3, we propose an OOD data synthesis strategy to adaptively select the hyperparameter $N$. Since the synthesized OOD data are derived from in-distribution (ID) information, they may not fully capture the characteristics of real-world OOD data. Nevertheless, our empirical findings indicate that such synthetic samples are sufficiently informative for selecting a robust hyperparameter $N$. As shown in Table 16, the optimal $N$ determined using synthetic OOD samples (*i.e.*, $N_{\text{syn}}$) is highly consistent with the one obtained from multiple real OOD datasets (*i.e.*, $N_{\text{real}}$), achieving comparable detection performance. These results validate the practicality of the $N$-selection strategy, even in the absence of real OOD data.

### E.4 The Impact of Hyperparameter N on LogitGap Performance

In Table 17, we provide a more ablation study on hyparameter $N$. Specifically, we conduct experiments under the zero-shot OOD detection setting, using either ImageNet-100 or ImageNet-1K as ID dataset. For simplicity, we report the FPR95 of our LogitGap method based on CLIP ViT-B/16 model. We can observe that: (1) Optimal $N$ varies by dataset (e.g., 19 for ImageNet-100, 195 for ImageNet-1K); (2) Setting $N$ to 20% of total classes consistently provides strong performance. Therefore, we adopt this value as default in LogitGap.

Table 17: Effect of hyperparameter $N$ on FPR95 using ViT-B/16 under zero-shot setting.

| | 5 | 95 | 195 | 295 | 395 | 495 | 595 | 695 | 795 | 895 | 995 |
|---|---|---|---|---|---|---|---|---|---|---|---|
| NINCO | 83.26 | 77.34 | 76.81 | 76.56 | 76.40 | 76.74 | 77.00 | 77.53 | 78.29 | 78.68 | 79.65 |
| ImageNet-O | 81.80 | 71.85 | 72.45 | 73.20 | 73.40 | 73.55 | 73.95 | 74.30 | 74.85 | 75.50 | 75.90 |
| ImageNetOOD | 82.49 | 75.47 | 76.38 | 77.16 | 77.64 | 78.26 | 78.74 | 79.19 | 79.72 | 80.31 | 81.04 |

(a) ImageNet as ID dataset.

| | 1 | 9 | 19 | 29 | 39 | 49 | 59 | 69 | 79 | 89 | 99 |
|---|---|---|---|---|---|---|---|---|---|---|---|
| NINCO | 75.54 | 46.27 | 42.77 | 41.88 | 41.85 | 42.6 | 43.23 | 44.49 | 46.31 | 47.64 | 50.58 |
| ImageNet-O | 76.00 | 45.70 | 43.50 | 44.25 | 44.30 | 45.25 | 45.75 | 46.45 | 47.90 | 47.90 | 48.65 |
| ImageNetOOD | 76.02 | 46.86 | 46.19 | 46.62 | 47.10 | 47.80 | 48.33 | 49.32 | 50.51 | 50.57 | 51.53 |

(b) ImageNet-100 as ID dataset.

