# OpenReview forum: "Revisiting Logit Distributions for Reliable Out-of-Distribution Detection"
_NeurIPS.cc/2025/Conference — NeurIPS 2025 poster_

### Official Review · Reviewer_AQ8y · 2025-06-23

**Clarity:** 4
**Significance:** 2
**Originality:** 2
**Rating:** 4
**Confidence:** 4

**Summary:**

This paper addresses the under-utilization of the logit space in post-hoc OOD detection by proposing the LogitGap scoring function and its focused subset variant (LogitGap-topN). The main idea is to explicitly measure the average gap between the maximum logit and the other logits, and to improve separability between ID and OOD samples by retaining only the top-N logits to filter out noisy tail values.

**Questions:**

How well do synthetic OOD samples align with real OOD distributions, and how does this affect performance?

**Ethical Concerns:**

["NO or VERY MINOR ethics concerns only"]

**Final Justification:**

Most of my concerns have been resolved. Once the authors add the summary of related methods from the rebuttal and clarify the limitations about ImageNet-R in the final version, I believe the paper's overall contribution will meet the borderline for acceptance. However, these limitations prevent me from giving a higher score.

**Limitations:**

yes

**Paper Formatting Concerns:**

no formatting issues noticed.

**Quality:**

2

**Strengths And Weaknesses:**

**strengths**

- The writing is very clear, articulating the motivation and method clearly
- The method is simple and easy to use

**weaknesses**

- The biggest issue may be that the method itself is very trivial and lacks novelty. Research around logit magnitudes can be traced back to the last century, and the concept of logit gap proposed in this paper has similar concepts that were introduced as early as 2001 (Margin Sampling) [1], and has been discussed in many subsequent works. Therefore, the method and insights proposed in this paper are not novel.
- The comparisons in the method section are not comprehensive enough. The zero-shot comparison targets in Tables 1 and 2 are all works from three years ago; while the effects when combined with other methods in Table 4 are often negative. In Table 5, the paper did not include several top-ranking works from the OpenOOD leaderboard for comparison, only comparing against methods with poorer performance.
- The ablation study on hyperparameter N with only one figure (Fig 3) is insufficient. It would be better to provide more comprehensive data. As stated in the paper, N has a significant impact on logitgap performance, and it clearly needs different tuning for different datasets and models. Although the authors provide a synthetic OOD data scheme, whether synthetic OOD data aligns with real OOD distributions, and the impact of the alignment degree on final performance, need further discussion.

**minor weaknesses**

- The title of Table 2 in the supplementary material is incorrect

[1] Tong, Simon, and Daphne Koller. "Support vector machine active learning with applications to text classification." *Journal of machine learning research* 2.Nov (2001): 45-66.

---

> ### Author Rebuttal · Authors · 2025-07-31
>
> ### **W1: The biggest issue may be that the method itself is very trivial and lacks novelty. Research around logit magnitudes can be traced back to the last century, and the concept of logit gap proposed in this paper has similar concepts that were introduced as early as 2001 (Margin Sampling).**
>
> We agree that the general idea of utilizing logits has a long history, our method makes a novel contribution in the specific context of OOD detection:
>
> - **Revisit Logit Distribution in OOD Detection.** Traditional methods (e.g. MaxLogit and MSP) use only the maximum logit or probability as OOD score, overlooking the rich information in non-maximum logits.  Recent Energy-based methods instead link high energy to low likelihood, but they struggle in zero-shot (CLIP-based) settings and fail to explicitly model the logit distribution. In contrast, our method revisits the full logit distribution and proposes a novel logit-gap-based scoring function that better distinguishes ID from OOD samples.
> -  **Different Context and Objective.** While Margin Sampling also involves the concept of logit gap, it was designed for active learning to identify uncertain samples for labeling. In contrast, our LogitGap leverages the logit gap to score and detect OOD samples, particularly from unseen classes or styles. To our knowledge, this is the first work to adapt the logit gap specifically for OOD detection in zero- and few-shot settings.
> -   **Simple but effective.**  Although the proposed method is conceptually simple, it consistently demonstrates strong performance across various OOD detection tasks, including zero-shot, few-shot, hard and traditional settings.
>
> -----
>
>
>
> ### **W2: The comparisons in the method section are not comprehensive enough.**
>
> ### **W2.1: Zero-shot comparison baselines in Tables 1 and 2 are from three years ago.**
>
> In response, we have added two recent zero-shot OOD detection methods [1,2]. See response to **reviewer L4Ge's Weakness 2.1(W2.1)** for details.
>
> - **Training-free, post-hoc simplicity**: Many recent zero-shot OOD methods require additional training data [3,4] or external models [5]. Differently, LogitGap is a simple, training-free, post-hoc method directly applicable to pre-trained CLIP models.
> -  **Specialized for vision-language models**: Most post-hoc OOD methods target unimodal visual backbones (e.g., ResNet), and often struggle with vision-language models like CLIP,  underscoring the need for our specialized approach.
>
> ### **W2.2: Effectiveness when combined with other methods in Table 4 are negative.**
>
> Table 4 evaluates the compatibility of LogitGap with existing fine-tuning methods. By simply replacing the OOD score, LogitGap improves performance on **9 out of 12** average metrics. The largest performance drop is under **0.2%**, indicating minimal downside. These results suggest that LogitGap can serve as a drop-in replacement in existing OOD frameworks, offering consistent gains with negligible risk.
>
>
> ### **W2.3: Comparison in Table 5 did not include several top-ranking works from the OpenOOD.**
>
> + **Training-Free & Data-Free Advantage**: Unlike top OpenOOD methods (e.g., OE-based approaches) that require large-scale data and training, LogitGap operates in a fully training-free and data-free manner.
> +  **Fair Comparison with Post-Hoc Baselines**: We benchmark against training-free post-hoc methods (e.g., SHE, NCI) in Table 5, where LogitGap shows superior performance.
> +  **Compatibility with Training-Based Methods**: LogitGap can also enhance training-based approaches—when combined with OE, performance further improves, demonstrating its versatility in OOD detection pipelines.
>
> -----
>
>
>
>
> ### **W3.1: The ablation study on hyperparameter N with only one figure (Fig 3) is insufficient. It would be better to provide more comprehensive data.**
>
>   In below table, we provide a more ablation study on hyparameter $N$. Specifically, we conduct experiments under the zero-shot OOD detection setting, using either ImageNet-100 or ImageNet-1K as ID dataset. For simplicity, we report the FPR95 of our LogitGap method based on CLIP ViT-B/16 model. We can observe that: (1) Optimal  $N$ varies by dataset (e.g., 19 for ImageNet-100, 195 for ImageNet-1K); (2) Setting $N$ to 20% of total classes consistently provides strong performance. Therefore, we adopt this value as default in LogitGap.
>
> | ImageNet-100: N | 1     | 9     | 19    | 29    | 39    | 49    | 59    | 69    | 79    | 89    |
> | --------------- | ----- | ----- | ----- | ----- | ----- | ----- | ----- | ----- | ----- | ----- |
> | ImageNetOOD | 76.02 | 46.86 | 46.19 | 46.62 | 47.10 | 47.80 | 48.33 | 49.32 | 50.51 | 50.57 |
> | ImageNet-O  | 76.00 | 45.70 | 43.50 | 44.25 | 44.30 | 45.25 | 45.75 | 46.45 | 47.90 | 47.90 |
>
> | ImageNet-1k: N  | 5     | 95    | 195   | 295   | 395   | 495   | 595   | 695   | 795   | 895   |
> | --------------- | ----- | ----- | ----- | ----- | ----- | ----- | ----- | ----- | ----- | ----- |
> | ninco       | 83.26 | 77.34 | 76.81 | 76.56 | 76.40 | 76.74 | 77.00 | 77.53 | 78.29 | 78.68 |
> | ImageNet-O  | 81.80 | 71.85 | 72.45 | 73.20 | 73.40 | 73.55 | 73.95 | 74.30 | 74.85 | 75.50 |
> | ImageNetOOD | 82.49 | 75.47 | 76.38 | 77.16 | 77.64 | 78.26 | 78.74 | 79.19 | 79.72 | 80.31 |
>
> ### **W3.2: Whether synthetic OOD data aligns with real OOD distributions, and the impact of the alignment degree on final performance, need further discussion.**
>
> - As stated in the paper, we propose a synthetic OOD sampling scheme to adaptively select $N$. Importantly, LogitGap also performs well **without synthetic OOD**, by simply setting $N$ to a fixed proportion (e.g., 20% of the number of classes).
> - We observe that while synthetic data may not fully capture real OOD characteristics, it is effective for selecting a suitable $N$ for LogitGap*. Please see our response to **Reviewer BQ1B's Weakness 3 (W3)** for more details.
>
> -----
>
>
>
> ### **Minor W1. The title of Table 2 in the supplementary material is incorrect.**
>
> Thank you for pointing this. We will fix it.
>
>
>
> [1] GL-MCM: Global and Local Maximum Concept Matching for Zero-Shot Out-of-Distribution Detection, IJCV 2025.
>
> [2] TAG: Text Prompt Augmentation for Zero-Shot Out-of-Distribution Detection, ECCV 2024.
>
> [3] CLIPN for Zero-Shot OOD Detection: Teaching CLIP to Say No, ICCV 2023.
>
> [4] Zero-Shot Out-of-Distribution Detection Based on the Pre-trained Model CLIP, AAAI 2022.
>
> [5] Envisioning Outlier Exposure by Large Language Models for Out-of-Distribution Detection, ICML 2024.

---

> > ### Comment · Reviewer_AQ8y · 2025-08-04
> >
> > Dear  Authors,
> >
> > Thank you for addressing my concerns.
> >
> > 1. I appreciate the rebuttal of W2.1, W2.3, W3.1, and W3.2.
> >
> > 2. For W1, I still believe a table summarizing related research on logit patterns is needed, given the long history of such methods.
> >
> > 3. Regarding W2.2, I remain concerned. While LogitGap shows average improvement in 9 of 12 comparisons, in the four-shot evaluation, it only improved in 10 of 18 cases, with 7 instances of performance decline. The authors should clarify limitations related to fine-tuning in the main text.
> >
> > Having addressed these two concerns, I would be open to raising my score.

---

> > > ### Author Response · Authors · 2025-08-06
> > >
> > > ### **D1. For W1, I still believe a table summarizing related research on logit patterns is needed, given the long history of such methods.**
> > >
> > > |**Method** (Year) | Task  | Equation|
> > > | - |- |- |
> > > | MSP (2022)| OOD Detection   | $\max_c p_c$|
> > > | MaxLogit (2022)   | OOD Detection   | $\max_c z_c$|
> > > | DML (2023) [4] | OOD Detection   | $\max_c \lambda \hat z_c+\|\|z_c\|\|$, $z_c=\hat z_c\cdot\|\|z_c\|\|$ |
> > > | GEN (2023) | OOD Detection   | $-\sum_{c=1}^M p^\gamma_{c} (1 - p_{c})^\gamma$, and $ p_{1}\geq \cdot\cdot\cdot \geq p_{M} \geq \cdot\cdot\cdot \geq p_{N} , \gamma \in (0,1)$ |
> > > | **Margin Sampling** (2006) [2] | Active Learning | $  p_1 - p_2$ and $p_1 \geq p_2 ... \geq p_N$ |
> > > | LC (2005) [1]| Active Learning | $\max_c 1- p_{c}$ |
> > > | Entropy (2008) [3] | Active Learning | $-\sum_cp_c\cdot \log p_c$   |
> > > | **LogitGap**| OOD Detection   | $\frac 1{M-1} \sum_{c=2}^M{z_1 - z_c}$ and  $z_1\geq \cdot\cdot\cdot\geq z_M \geq \cdot\cdot\cdot\geq z_N$ |
> > >
> > > Thank you for the reviewer's continued feedback.
> > > We have reviewed related works in both active learning and OOD detection, and summarized several representative methods in the above table.
> > > Here, $z_c$ denotes the logit and $p_c$ represents the predicted probability for $c$-th class.
> > >
> > > Specifically, we compare our **LogitGap** with three classic active learning approaches: Least Confidence (LC) [1], Margin Sampling [2], and Maximum Entropy (Entropy) [3]; and four classic OOD detection methods: MSP, MaxLogit, Decoupling MaxLogit (DML) [4] and Generalized Entropy (GEN).
> > > There are some recent methods in active learning, such as Query-by-Committee (QBC) [5] and Gradient Embedding [6]. We do not include them as these approaches differ significantly from ours in methodology and assumptions.
> > >
> > > Among them, Margin Sampling is the most relevant to ours, as it also computes the gap in model outputs.
> > > However, there are **two differences** between Margin Sampling and our LogitGap:
> > >
> > > - Margin Sampling operates on predicted probabilities, whereas LogitGap use raw logits;
> > > - Margin Sampling considers only the top-2 outputs,
> > > while LogitGap uses the top-M.
> > >
> > > To highlight the importance of these differences, we adapt Margin Sampling  for OOD detection, under a
> > >  zero-shot  setting using CLIP ViT-B/16, with ImageNet categories as ID and categories from other benchmarks as OOD classes. As shown in the table below,  LogitGap consistently outperforms Margin Sampling across all benchmarks, demonstrating the effectiveness of our approach.
> > >
> > >
> > > | ImageNet| | ninco | | | ImageNet-O | || ImageNetOOD | | | AVG   | |
> > > | - | - | - | - | - | -- | - | -|- | - | - | - | - |
> > > |method|FPR95 ↓|AUROC ↑|FPR95 ↓|AUROC ↑|FPR95 ↓|AUROC ↑|FPR95 ↓|AUROC ↑|FPR95 ↓|AUROC ↑|FPR95 ↓|AUROC ↑ |
> > > | Margin Sampling | 89.88 | 68.40  | 94.97 | 89.45 | 67.83  | 98.14 | 89.96| 67.49   | 77.67 | 89.76 | 67.91 | 90.26 |
> > > | LogitGap| **77.42** | **76.51** | **96.38** | **71.95** | **81.45**  | **99.03** | **75.40** | **80.27**   | **86.21** | **74.92** | **79.41** | **93.87** |
> > >
> > > [1] Reducing labeling effort for structured prediction tasks, AAAI 2005
> > >
> > > [2] Margin-based active learning for structured output spaces, ECML 2006
> > >
> > > [3] An Analysis of Active Learning Strategies for Sequence Labeling Tasks, EMNLP 2008
> > >
> > > [4] Decoupling MaxLogit for Out-of-Distribution Detection, CVPR 2023
> > >
> > > [5]  Query-by-committee improvement with diversity and density in batch active learning, Information Sciences 2018.
> > >
> > > [6] Deep batch active learning by diverse, uncertain gradient lower bounds, ICLR 2020
> > >
> > >
> > >
> > > ### **D2: In the four-shot evaluation, LogitGap improves performance in 10 of 18 cases, with 7 instances of performance decline. The authors should clarify limitations related to fine-tuning in the main text.**
> > >
> > > Thank you for pointing out this problem.
> > > We find that the performance decline primarily occurs on ImageNet-R.
> > > To further investigate this, we conduct experiments under 8-shot and 16-shot OOD detection settings, using ImageNet as the ID dataset and ImageNet-R as the OOD dataset.
> > >
> > > As shown in the table below, we observe that the performance decline gradually diminishes as the number of shots increases.
> > > For example, LogitGap reduces the FPR of CoOp by 1.21% in the 4-shot setting, but only by 0.31% in the 8-shot setting. The exact cause requires further study. We will clarify this limitation in the revised manuscript and discuss it further.
> > >
> > >
> > > | ImageNet|| ImageNet-R| |
> > > |-|-|-|-|
> > > | Method  | FPR95 $\downarrow$| AUROC $\uparrow$  | AUPR $\uparrow$|
> > > | **4-shot**  ||| |
> > > | CoOp| 78.38 | 73.53 | 92.02  |
> > > | +LogitGap   | 79.59  (**$\downarrow$ 1.21**) | 71.94 (**$\downarrow$ 1.59**)  | 91.29  (**$\downarrow$  0.73**) |
> > > | **8-shot**  ||| |
> > > | CoOp| 76.21 | 74.49 | 92.12  |
> > > | +LogitGap   | 76.52 (**$\downarrow$ 0.31**)  | 74.10  (**$\downarrow$ 0.39**) | 91.89  (**$\downarrow$ 0.23**)  |
> > > | **16-shot** ||| |
> > > | CoOp| 78.95 | 73.62 | 91.98  |
> > > | +LogitGap   | 78.86 ($\uparrow$ **0.09**)| 73.29 (**$\downarrow$ 0.33**)  | 91.75 (**$\downarrow$ 0.23**)   |

---

> > > > ### Comment · Reviewer_AQ8y · 2025-08-06
> > > >
> > > > Dear Authors,
> > > >
> > > > Thank you for addressing my concerns. I appreciate your summary and discussion on W2.2. I will raise my score. Please ensure that all the results and tables from the rebuttal, especially regarding W1, are included in the final version.

---

> > > > > ### Author Response · Authors · 2025-08-08
> > > > > **Official Comment by Authors**
> > > > >
> > > > > Thank you again for your kind response and for taking the time to review our rebuttal. We truly appreciate your constructive feedback in helping to improve our work.

---

### Official Review · Reviewer_L4Ge · 2025-06-30

**Clarity:** 2
**Significance:** 2
**Originality:** 2
**Rating:** 4
**Confidence:** 5

**Summary:**

This work proposes a post-hoc OOD score based on the gap between the maximum logit and the remaining logits. Specifically, they focus on the Top-$N$ logits,  the threshold $N$ is computed using only a small number of ID samples.  If ID samples are not available, $N$ is set to be 20% of the total number of classes $K$.

**Questions:**

See weaknesses.

**Ethical Concerns:**

["NO or VERY MINOR ethics concerns only"]

**Final Justification:**

During the rebuttal, the authors have addressed most of my initial concerns, and I am willing to increase my score.
However, I believe the authors should revise the manuscript to better contextualize the contribution of LogitGap and include additional experiments to support the claims in the paper, e.g., ... achieves state-of-the-art performance .... .

Based on the rebuttal results,  LogitGap achieves performance comparable to GEN without access to ID images. When ID images are available, LogitGap combined with Residual achieves outperforms ViM. In that case, GEN combined with Residual should also be evaluated for a fair comparison. My concern is that the manuscript overstates the performance of LogiGap due to incomplete comparisons.

I believe the main contribution of LogitGap lies in its effectiveness on zero-shot and few-shot OOD detection (using CLIP-style models). Therefore,  the title and abstract should be revised to reflect this main message more accurately.

**Limitations:**

yes

**Quality:**

2

**Strengths And Weaknesses:**

Strengths:
- The idea is simple and effective compared to Maxi-Logit.
- It demonstrates the performance of LogitGap on both semantic-shifted OOD detection and covariate-shifted OOD detection.
- Experiments on semantic-shift OOD detection are conducted under both zero-shot and few-shot settings.

Weaknesses:
- The introduction section is poorly written.
    - Line 27-18, I would say energy score is the most representative OOD score because of its simplicity and effectiveness compared to MSP score.
    -  Line 32-33, it is true that MCM/MSP lacks information from other logits.  However, it's abrupt that the authors simply ignore the energy score (using full logit information) and directly raise the question of utilizing information from non-maximum logits. Could the authors clarify their reasoning for overlooking the energy score and elaborate on their motivation at this point?
    - Line 38-40, although the authors make an effort to demonstrate the effectiveness of LogiGap in Figure 1, the caption does not provide any information about the datasets or models used. Additionally, I believe the authors should include energy score for comparison in Figure 1.


- The experiments are not sufficient.
    -  For zero-shot OOD detection setting, two baselines are missing including GL-MCM [1] and TAG [2]. Meanwhile, the performed   OOD benchmark is not standard. Specifically, the evaluation of far-OOD detection is missing.  See https://zjysteven.github.io/OpenOOD/ for complete evaluation.
    -  For traditional OOD detection, two baselines are missing in the experiments section including ViM (using information from logit and feature) and GEN [3] (using information from probability space). Meanwhile, is there any reason that the authors switch from the large OOD benchmark (ImageNet-1k as ID dataset) to small OOD benchmark (CIFAR-10 as ID dataset)? Additionally, the results across different models such as ViT and BiT are also expected to present.
    - All results are only demonstrated in CLIP ViT-B-/16, which fails to show the generalization of the proposed method across different models. If the authors focus on the vision-language models,  it is expected to evaluate on the other ViT models including ViT-B/32 and ViT-L/14 and ResNet-based models including ResNet-50 and ResNet-101.
    - To support the claim in Line 159 -161, a more comprehensive empirical evaluation across different ID and OOD datasets with different models should be done.


Minor weaknesses:
- What datasets and models are utilized to produce Figure 2?
- Line 50-51, the concept of focusing on the top-N logits is also mentioned in GEN [1], which demonstrates that using the energy score with top 10% of logit values improves OOD performance.  Again, the author probably could try to compare the performance of LogitGap with the energy score using top 10% logit information.
- Based on the results in Table 3, specifically the FPR95 and AUROC metrics, it does not appear that OOD detection on ImageNet-10 and ImageNet-20 is particularly challenging, as the FPR95 values are close to 0 and the AUROC values are near 100.


[1] GL-MCM: Global and Local Maximum Concept Matching for Zero-Shot Out-of-Distribution Detection, 2023.

[2] TAG: Text Prompt Augmentation for Zero-Shot Out-of-Distribution Detection, 2024.

[3] GEN: Pushing the Limits of Softmax-Based Out-of-Distribution Detection, 2023.

---

> ### Author Rebuttal · Authors · 2025-07-31
>
> ### W1: The introduction section is poorly written.
>
> **W1.1&1.2: Energy-based method is the most representative OOD score, which also utilizes information from non-maximum logits. Could the authors clarify their reasoning for overlooking this method?**
>
> We would like to clarify that the energy-based method differs significantly from the other methods mentioned in the introduction, such as MaxLogit, MSP, and LogitGap. The energy-based method distinguishes between ID and OOD samples based on **likelihood density**, with the idea that high-energy points correspond to low likelihood density, making them more likely to be OOD samples. In contrast, the other methods focus on the **logit pattern**, with the idea that ID samples typically produce more peaked logits, while OOD samples corresponds to more uniform logit patterns. For this reason, we chose not to discuss energy-based methods in the introduction.
> In the experments, we **have compared energy-based methods with our LogitGap**, and results show that LogitGap outperforms the energy score across multiple benchmarks (see Table 1, 2 and 3).
> In addition, while energy-based method utilizes the non-maximum logits, we observe notable limitations of it from experimental results:
>
> - Energy-based method is **not compatible with zero-shot OOD scenarios** (see Tables 1, 2, and 3) when using a pre-trained CLIP model.
> We attribute this to different training losses: CLIP model is trained using an image-text contrastive loss, which differs significantly from the NLL loss commonly assumed in energy-based methods.
> As a result, energy-based methods perform poorly in this setting, since high energy values do not necessarily correspond to low likelihood densities during inference.
> - The log-sum-exp formulation is **numerically dominated by the largest logit**, which makes it behave similarly to MaxLogit in many cases.
>
> **W1.3: The caption of Figure 1 lacks information about datasets and models used. Additionally, the comparsion with energy-based method should be included in Figure 1.**
>
> Thanks to point out this problem. The results in Figure 1 are reported on the zero-shot OOD scenario with CLIP ViT-B/16 model. In this experiment, categories from ImageNet-20 are used as ID, while those from ImageNet-10 are used as OOD.
> For the comparison with energy-based method in Figure 1, we will revise Figure 1 in our updated version.
>
> ### W2: The experiments are not sufficient.
>
> **W2.1: Missing baselines like GL-MCM and TAG in zero-shot OOD detection.**
>
> We report the results of GL-MCM and TAG in the table below.
> The experiments are conducted using CLIP ViT-B/16, with ImageNet categories as ID classes and categories from other benchmarks as OOD classes.
> As shown in Table 2.1, LogitGap achieves comparable or better performance than the two methods on average across multiple benchmarks.
>
> |Table 2.1|ninco||ImageNet-O ||ImageNetOOD ||ssb_hard  |||AVG||
> |-|-|-|-|--|-|-|-|--|-|-|-|
> |Method|FPR95|AUROC|FPR95|AUROC|FPR95|AUROC|FPR95|AUROC|FPR95↓|AUROC↑|AUPR ↑ |
> |Energy|84.1|72.0|81.6|75.6|79.1|76.8|91.3|59.6|84.0|71.0|84.1|
> |+TAG|83.1|71.2|79.7|77.1|78.3|78.0|93.9|57.3|83.8|70.9|84.3|
> |MCM|79.7|73.6|75.9|79.5|81.0|78.3|89.5|63.8|81.5|73.8|85.9|
> |+TAG|81.6|71.3|77.6|79.9|83.0|78.8|90.6|63.3|83.2|73.3|86.1|
> |MaxLogit|79.4|74.4|77.2|77.9|75.9|78.7|89.6|60.7|80.5|72.9|84.7|
> |+TAG|77.7|73.9|74.5|79.5|**75.2**|80.1|91.6|59.1|79.8|73.2|85.1|
> |GL-MCM|**74.4**|76.0|72.4|79.5|79.2|77.3|**87.4**|**66.1**|78.3|74.7|86.2|
> |LogitGap|77.4|**76.5**|**71.9**|**81.5**|75.4|**80.3**|88.2|64.5|**78.2**|**75.7**|**86.4**|
>
> **W2.2: The performed OOD benchmark is not standard. Evaluation for far-OOD detection is missing.**
>
> Our benchmark design is inspired by recent observations [1], which highlight that **existing OOD datasets often mix covariate and semantic shifts**, making it difficult to isolate and accurately measure a model’s ability to detect true *semantic shift*. To address this, we construct a more challenging benchmark using NINCO, ImageNet-O, and ImageNet-OOD to emphasize **semantic shift** under a high-coverage, diverse, and realistic setting.
>
> Moreover, we have followed the OpenOOD protocol and added results on the far OOD scenario in Table 2.2 and included a comparison with the SSB_Hard dataset in Table 2.1 with ImageNet as ID set. These additions confirm LogitGap's robustness under a broader distributional shift.
>
> |Table2.2|iNaturalist||Textures||OpenImage-O |||AVG||
> |-|-|-|-|-|-|-|-|-|-|
> ||FPR95|AUROC|FPR95|AUROC|FPR95|AUROC|FPR95↓|AUROC↑|AUPR↑|
> |Energy|73.9|87.1|92.7|66.1|65.6|85.91|77.4|79.7|95.5|
> |MCM|31.5|94.4|58.8|85.8|40.8|91.99|43.7|90.7|97.9|
> |MaxLogit|56.3|90.5|86.5|71.9|51.9|88.93|64.9|83.8|96.4|
> |LogitGap|**27.8**|**94.9**|**58.2**|**85.7**|**37.3**|**92.7**|**41.1**|**91.1**|**98.0**|
>
> **W2.3: Missing baselines like ViM and GEN in traditional OOD detection.**
>
> We report the results of ViM and GEN in the table below.
> The experiments are conducted in traditional OOD detection setting, using CIFAR-10 as ID dataset.
> Specifically, since VIM requires training set statistics, we randomly selected 100 samples from the training set for estimation. In Table 2.3, LogitGap remains competitive, further validating its general applicability.
>
> |Table2.3|Near|OOD|Far|OOD|
> |-|-|-|-|-|
> |method|FPR95 ↓|AUROC ↑|FPR95 ↓|AUROC ↑|
> |ViM|68.7|71.4|56.7|75.5|
> |GEN|53.7|88.2|34.7|91.4|
> |Energy|61.3|87.6|41.7|91.2|
> |Energy+top10%|61.3|87.5|41.7|91.1|
> |LogitGap|**44.5**|**88.9**|**30.1**|**91.5** |
>
> **W2.4: Is there any reason that the authors switch from the large OOD benchmark (ImageNet-1k as ID dataset) to small OOD benchmark (CIFAR-10 as ID dataset)?**
>
> Thank you for this question.
> We report results on the small-scale OOD benchmark to demonstrate the generality of our method.
> We also conduct experiments on a larger OOD benchmark (CIFAR-100 as ID dataset) in the Appendix C.5.
> Furthermore, we additionally report results on ImageNet-1k with ResNet50 under the traditional OOD setting in following table, using SSB-Hard and NINCO as Near-OOD datasets, and iNaturalist, Textures, and OpenImage-O as Far-OOD datasets. Due to space limitations, we report the average performance of the method across all datasets.
> As shown in the table, our LogitGap achieves comparable or better performance than other methods across different benchmarks, and obtains the best average performance overall.
>
> |Table2.4|AVG||
> |--|--|-|
> |method|FPR95|AUROC|
> |MSP|49.1|84.6|
> |MaxLogit|50.1|84.3|
> |KLM |57.0|83.2|
> |ODIN|55.4|83.6|
> |Energy|50.5|84.0|
> |Energy+top10%|50.5|84.0|
> |LogitGap|**47.6**|**84.7**|
>
> **W2.5: For traditional OOD detection, the results across different models such as ViT and BiT are also expected to present.**
>
> Here, we report additional results in traditional OOD setting with different models, including Swin-T (Table 2.5.1) and ViT-B/16 (Table 2.5.2), to highlight the method's generalization.
> In experiments, we utilize ImageNet-1k as ID dataset and other benchmarks as OOD datasets.
> As shown in Table 2.5.1 and 2.5.2, our LogitGap achieves better performance than other comparative methods across different models.
>
> |Table2.5.1|Near|OOD|Far|OOD|
> |-|--|-|-|-|
> |method|FPR95 ↓|AUROC ↑|FPR95 ↓|AUROC ↑|
> |MSP|76.1|76.6|59.3|86.4|
> |MaxLogit|77.3|75.7|67.8|84.8|
> |Energy|78.3|73.2|75.6|81.3|
> |LogitGap|**69.6**|**78.2**|**39.1**|**89.7**|
>
> |Table2.5.2|Near|OOD|Far|OOD|
> |-|-|-|-|-|
> |method|FPR95 ↓|AUROC ↑|FPR95 ↓|AUROC ↑|
> |MSP|90.7|70.6|68.3|85.5|
> |MaxLogit|92.3|68.3|79.2|83.5|
> |Energy|93.2|62.4|85.3|79.0|
> |LogitGap|**85.0**|**72.7**|**47.6**|**88.6**|
>
> **W2.6: For zero-shot OOD detection, it is expected to evaluate on the other CLIP variants, including ViT models and ResNet models.**
>
> We extend our experiments to other CLIP variants including ViT-L/14 and ResNet-101. Results in Table 2.6 demonstrate that LogitGap generalizes well across architectures. As shown in the table below, our method consistently outperforms baselines across all backbones, indicating strong generalization. These results have been included in the revised version of the paper.
>
> |Table2.6|Near|OOD|Far|OOD|
> |-|-|-|-|-|
> |**ViT-L/14**|FPR95↓|AUROC↑|FPR95↓|AUROC↑|
> |Energy|82.0|73.9|76.5|79.2|
> |MCM|74.4|77.7|41.1|90.8|
> |MaxLogit|77.1|75.9|63.2|83.2|
> |LogitGap|**68.9**|**80.3**|**38.0**|**91.6**|
> |**RN-101**|||||
> |Energy|88.1|67.9|94.2|69.6|
> |MCM|85.9|70.3|58.0|87.4|
> |MaxLogit|85.9|69.9|86.8|75.5|
> |LogitGap|**84.9**|**71.9**|**57.5**|**87.4**|
>
> **W2.7: To support the claim in Line 159 -161, a more comprehensive evaluation across different ID/OOD datasets with different models should be done.**
>
> To further support our claim, we report statistical information on the logits distribution across multiple datasets (**see our response to Reviewer 4oYE’s Comment W1**). These results show that the observed logit pattern is consistently present across diverse datasets.
>
> ### MW1: What datasets and models are utilized to produce Figure 2?
>
> Figure 2 was produced using the CLIP ViT-B/16 model, with ImageNet100 as the ID and iNaturalist as the OOD dataset.
>
> ### MW2: Compare LogitGap with the energy score using top 10% logit information.
>
> We have compared LogitGap with top-10% energy scoring and found LogitGap achieves consistently better results (see Table 2.3, 2.4), validating its stronger separability for OOD detection.
>
> ### MW3: Experimental setup on ImageNet-10/ImageNet-20 is not particularly challenging.
>
> We follow the experimental setup from previous works [2], alternately using ImageNet-10 and ImageNet-20 as the ID and OOD datasets to create a more challenging scenario.
> While existing methods achieve high AUROC scores in this setting, obtaining a low FPR95 remains difficult.
> For example, LogitGap reduces FPR95 by 8.40% compared to MCM, highlighting its advantage in handling subtle distribution shifts.
>
> [1] IMAGENET-OOD: Deciphering Modern Out-of- Distribution Detection Algorithms, ICLR 24
>
> [2] Delving into Out-of-Distribution Detection with Vision-Language Representations, NeurIPS 22

---

> > ### Comment · Reviewer_L4Ge · 2025-08-04
> >
> > Thanks for the detailed responses.
> > - W1.1 & W1.2: Thanks for the clarification. It is recommended that the authors revise the introduction accordingly.
> > - W1.3 is addressed.
> > - W2.1-2.2 are addressed. Thanks for conducting the experiments!
> > -  W2.3 are addressed.
> > -  W2.4 - 2.5 are partially addressed.  ViM and GEN baselines are included for CIFAR-100 in Table 2.3, but are missing from the ImageNet-1k results in Tables 2.4 and 2.5. Could the authors clarify the reason for their omission?
> > -  W2.6 -W2.7 are addressed.

---

> > > ### Author Response · Authors · 2025-08-06
> > >
> > > ### **D1: It is recommended that the authors revise the introduction accordingly.**
> > >
> > > We sincerely appreciate your feedback. We will revise the relevant part of the Introduction in the revised manuscript accordingly.
> > >
> > >
> > >
> > > ### **D2: ViM and GEN baselines are included for CIFAR-100 in Table 2.3, but are missing from the ImageNet-1k results in Tables 2.4 and 2.5. Could the authors clarify the reason for their omission?**
> > >
> > > Thanks for pointing out this problem.
> > > To provide a more comprehensive comparison, we additionally conduct experiments on ImageNet-1K.
> > > Due to time limits, we perform traditional OOD detection using the ResNet-50 model, with ImageNet-1k as ID dataset and other benchmarks as OOD datasets
> > > The results are presented in the table below.
> > >
> > > |          | Near OOD |         |         |         |             |         | Far OOD |         |             |         |         |         |
> > >   | -------- | -------- | ------- | ------- | ------- | ----------- | ------- | ------- | ------- | ----------- | ------- | ------- | ------- |
> > >   | method   | ssb_hard |         | ninco   |         | iNaturalist |         | TEXTURE |         | OpenImage-O |         | **AVG**     |         |
> > >   |          | FPR95 ↓  | AUROC ↑ | FPR95 ↓ | AUROC ↑ | FPR95 ↓     | AUROC ↑ | FPR95 ↓ | AUROC ↑ | FPR95 ↓     | AUROC ↑ | FPR95 ↓ | AUROC ↑ |
> > >   | GEN      | 75.71    | 72.01   | 54.88   | 81.7    | 26.11       | 92.44   | 46.24   | 87.59   | 34.52       | 89.26   | **47.49**   | 84.60   |
> > >   | VIM      | 98.18    | 33.67   | 95.64   | 37.71   | 96.02       | 30.24   | 92.46   | 53.72   | 96.54       | 36.11   | 95.77   | 38.29   |
> > >   | LogitGap | 75.48    | 72.51   | 53.64   | 82.25   | 26.68       | 92.24   | 47.11   | 87.15   | 35.10       | 89.11   | 47.60   | **84.65**   |
> > >
> > > As shown in the table, our LogitGap  outperforms ViM across different benchmarks and achieves performance comparable to GEN on average.
> > > These results demonstrate that LogitGap can match or surpass the performance of both methods with a large-scale OOD benchmark.

---

> > > > ### Comment · Reviewer_L4Ge · 2025-08-06
> > > >
> > > > Thanks for conducting the experiments. However, I suspect there may be implementation issues in your ViM reproduction, as the results you obtained are significantly worse than those reported in the OpenOOD benchmark (https://zjysteven.github.io/OpenOOD/). Furthermore, since ViM utilizes both logit feature information, it is generally expected to outperform GEN.
> > > >
> > > > Overall, LogitGap appears to be effective on small-scale OOD benchmarks (as shown in Table 2.3 of the rebuttal) and achieves competitive performance on large-scale benchmarks among vision-based post-hoc OOD detection methods. For zero-shot and few-shot OOD detection (using CLIP-style models), LogitGap can be regarded as an enhancing method such TAG to further improve performance (Tables 1 and 2 in the main paper). I recommend that the authors revise the manuscript to  better convey the main message of LogitGap.

---

> > > > > ### Author Response · Authors · 2025-08-07
> > > > >
> > > > > ### **Second-Round D1: I suspect there may be implementation issues in your ViM reproduction, as the results you obtained are significantly worse than those reported in the OpenOOD benchmark. Furthermore, since ViM utilizes both logit feature information, it is generally expected to outperform GEN.**
> > > > >
> > > > > Thank you very much for your reply. There may be a misunderstanding regarding the experimental results of ViM.
> > > > >
> > > > > First, as clarified in our response to W2.3, we reproduced ViM under a setting with only  **100 randomly selected ID samples** used for training, matching the setup used for LogitGap to ensure a fair comparison. As shown in the table below, ViM performs poorly with limited data. We argue that ViM relies on decomposing ID feature matrix, a small number of ID samples may fail to capture the true distribution, thus impairing OOD detection performance.
> > > > >
> > > > > To align with the OpenOOD benchmark results, we also conducted a experiment where **all ID training samples** were used to compute ViM. As shown in the table below, in this setting, ViM's performance aligns with OpenOOD benchmark results and surpasses GEN. Importantly, **LogitGap still performs competitively**, even surpassing ViM on 3 out of 5 datasets.
> > > > >
> > > > > Finally, since ViM benefits from feature space information, we introduced a **feature-enhanced variant** of LogitGap by combining it with Residual [1] (a method that captures compressed representations of training features used in ViM).  **LogitGap + Residual** achieves the **best average performance**, demonstrating that LogitGap can also be extended to benefit from feature-based cues in traditional OOD detection settings. We will include these experiments in the final manuscript.
> > > > >
> > > > > | ImageNet       | Near OOD  |  |  |  |    |  | Far OOD   |  |    |  |  |  |
> > > > > | ----------------------- | --------- | --------- | --------- | --------- | ----------- | --------- | --------- | --------- | ----------- | --------- | --------- | --------- |
> > > > > | method| ssb_hard  |  | ninco     |  | iNaturalist |  | TEXTURE   |  | OpenImage-O |  | AVG       |  |
> > > > > |       | FPR95 ↓   | AUROC ↑   | FPR95 ↓   | AUROC ↑   | FPR95 ↓     | AUROC ↑   | FPR95 ↓   | AUROC ↑   | FPR95 ↓     | AUROC ↑   | FPR95 ↓   | AUROC ↑   |
> > > > > | GEN   | 75.71     | 72.01     | 54.88     | 81.70      | 26.11       | 92.44     | 46.24     | 87.59     | 34.52       | 89.26     | 47.49     | 84.60     |
> > > > > | **100 ID samples**      |  |  |  |  |    |  |  |  |    |  |  |  |
> > > > > | ViM   | 98.18     | 33.67     | 95.64     | 37.71     | 96.02       | 30.24     | 92.46     | 53.72     | 96.54       | 36.11     | 95.77     | 38.29     |
> > > > > | _LogitGap_     | **75.48** | **72.51** | **53.64** | **82.25** | **26.68**   | **92.24** | **47.11** | **87.15** | **35.10**    | **89.11** | **47.60** | **84.65** |
> > > > > | **all ID samples**      |  |  |  |  |    |  |  |  |    |  |  |  |
> > > > > | ViM   | 80.41     | 65.54     | 62.29     | 78.63     | 30.67       | 89.56     | **10.49** | **97.97** | 32.82       | 90.50      | 43.34     | 84.44     |
> > > > > | _LogitGap_     | **75.25** | **72.73** | **52.61** | **82.61** | 26.93       | 92.15     | 47.32     | 86.81     | 35.57       | 88.98     | 47.54     | 84.66     |
> > > > > | _LogitGap+Residual_ [1] | 77.83     | 68.86     | 55.82     | 81.27     | **23.87**   | **92.46** | 19.12     | 96.30     | **29.49**   | **91.15** | **41.23** | **86.01** |
> > > > >
> > > > > [1] Out-of-distribution with virtual-logit matching. CVPR, 2022.
> > > > >
> > > > >
> > > > >
> > > > >
> > > > > ### **Second-Round D2:  LogitGap appears to be effective on small-scale OOD benchmarks and achieves competitive performance on large-scale benchmarks ..... I recommend that the authors revise the manuscript to better convey the main message of LogitGap.**
> > > > >
> > > > > Thank you for this valuable suggestion.
> > > > >
> > > > > - In the traditional OOD setting (Section 5.3, Lines 305-323), LogitGap achieves better performance on small-scale OOD benchmarks (Table 5 in the main paper and Table 2.3 in the rebuttal), and competitive performance on large-scale benchmarks (as shown in our response to W2.4 and D2).  We will include these experimental results and their discussion in section 5.3 of the revised version.
> > > > > - Notably, our work mainly focuses on zero-shot and few-shot OOD detection using a pre-trained CLIP model. In this setting, our LogitGap consistently outperforms existing methods across both small-scale benchmarks (Table 2 and Table 3 in the main paper) and large-scale benchmarks (Table 1 and Table 4 in the main paper).
> > > > > - Our LogitGap can be integrated with several few-shot OOD detection methods (Lines 252–257, 284–287) as well as traditional OOD detection approaches (Lines 319-323 and our response to Second-Round D1), and effectively improves the performance of these baseline methods across various benchmarks. Therefore, our method can be regarded as an enhancing method to further improve performance for several OOD methods. We will revise the introduction and conclusion to emphasize this point.
> > > > >
> > > > > We will convey these messages of LogitGap in the revised version.

---

### Official Review · Reviewer_4oYE · 2025-07-02

**Clarity:** 2
**Significance:** 3
**Originality:** 3
**Rating:** 4
**Confidence:** 4

**Summary:**

This paper proposes LogitGap, a novel scoring function for out-of-distribution (OOD) detection. LogitGap is designed to address the limitations of existing methods such as MaxLogit and Maximum Concept Matching (MCM). Specifically, while MaxLogit considers only the highest class logit and ignores the rest of the distribution, MCM relies on the softmax function, which normalizes the logit values and may obscure informative differences. LogitGap is formulated as the gap between the top logit and the rest of the distribution, thereby combining the advantages of both MaxLogit (absolute logit norm) and MCM (distributional awareness). The paper provides both theoretical insights and extensive empirical results to demonstrate the effectiveness of LogitGap.

**Questions:**

1. Why this specific LogitGap formulation?
The motivation to combine the strengths of MaxLogit and MCM is clearly explained, and the experimental results support the effectiveness of LogitGap. However, the specific choice of computing the difference between the maximum logit and the average of the remaining logits requires further justification. Why is this particular subtraction-based formulation considered the most effective? For example, were alternative designs such as weighting the MCM score by the max logit or using other aggregation strategies considered? Providing experimental comparisons or ablation study results on these alternatives would help strengthen the justification for the robustness or optimality of the proposed formulation.

2. Sensitivity to N (number of logits averaged):
The sensitivity of LogitGap to hyperparameter N is a significant concern for practical applications. While N averages non-maximum logits, low-ranked classes likely have consistently low values, regardless of whether they are in-distribution (ID) or out-of-distribution (OOD). This could reduce the average of the subtraction term, making ID/OOD distinction difficult. This challenge also affects MCM. We anticipate MCM would perform better if evaluated with an N value derived similarly to LogitGap's N. A comparison under such conditions would ensure a fair evaluation. We recommend conducting this experiment and sharing the results.

**Ethical Concerns:**

["NO or VERY MINOR ethics concerns only"]

**Final Justification:**

This paper proposes LogitGap, a simple yet practical and powerful algorithm for OOD detection. The authors justify their focus on the CLIP encoder setting by noting that existing energy-based methods underperform in this context.

Many reviewers initially pointed out areas for improvement in the introduction, figures, and theorems. However, the authors responded effectively with additional experiments that demonstrated the strengths of their methodology and presented a concrete plan for revision. Furthermore, they highlighted their method's impressive generality by showing strong performance on tasks beyond image classification, such as audio classification.

While there were initial concerns regarding the quality of the draft and the novelty of applying a classical method, the authors' thorough rebuttal and the clear empirical strengths of their work are highly convincing. The demonstrated effectiveness, the successful resolution of key issues, and the method's broad applicability are significant strengths. For these reasons, I have decided to raise my score from my initial rating.

**Limitations:**

1. **Motivation Example in Section 4.1 may be unintuitive**:
   In the provided example, 𝑧₂ shows high logits for two classes and is described as likely being an in-distribution sample, while 𝑧₁ lacks a dominant logit. While this may be intuitive from an OOD detection perspective, it contradicts typical classification expectations, where such ambiguity often indicates a confusing or hard-to-classify input. This could lead to confusion for readers. If a better illustrative example is not available, the current description should be revised for clarity. For instance, instead of:
   "Despite the significant difference in magnitude and sharpness..."
   a more intuitive phrasing might be:
   "Since OOD samples tend not to show confident alignment with any known class, a sample like 𝑧₂, which activates multiple classes relatively strongly, may indicate a higher likelihood of being in-distribution."

2. **Excessive reliance on supplementary material for theoretical explanation**:
   As previously mentioned, key derivations—especially Theorem 4.1—are presented only in the supplementary material. This is problematic given the central importance of this result. The main text does not provide sufficient intuition or explanation about why the theorem holds or how it contributes to the final formulation. Without such context, readers may struggle to assess the theoretical soundness of the method. Including a brief, self-contained summary or intuition for the theorem in the main paper would significantly improve clarity.

3. **Limited generalization beyond vision classification tasks**:
   Although the method is theoretically general, its evaluation is confined solely to vision classification benchmarks. Given the simplicity of the formulation and its potential for cross-domain applicability, including at least one experimental setting beyond vision classification would strengthen the paper. This would help demonstrate the broader applicability and robustness of the method across different domains.

**Paper Formatting Concerns:**

Nothing

**Quality:**

3

**Strengths And Weaknesses:**

Strengths:

1. The proposed method is conceptually simple and easy to understand, making it broadly applicable across various domains.

2. The authors support their approach with both comprehensive theoretical analysis and extensive experimental validation, which strengthens the credibility of the method.

3. The paper is well-organized and clearly written, facilitating accessibility and readability for a wide range of readers.

Weaknesses:

1. The theoretical motivation of the proposed method relies on certain assumptions that are not fully verified or supported. If these assumptions are trivial or well-established, proper references should be provided to increase the logical soundness. For instance, Figure 2 lacks critical context — it is unclear under what setting it was generated, whether it represents a single data point or an average over multiple samples.

2. The applicability of the method is demonstrated solely on vision classification tasks, which limits the scope of the contribution. Given the simplicity and generality of the proposed formulation, it would be more compelling to show its effectiveness across a broader set of domains or tasks.

3. The theoretical explanation is overly reliant on the supplementary material. While it is acceptable to place extended proofs in the appendix, the main text should still provide enough intuition and explanation for readers to understand the formulation without constantly referring to the supplement. As it stands, readers may find it difficult to follow the logic behind the proposed metric.

---

> ### Author Rebuttal · Authors · 2025-07-31
>
> ### **W1.1 The theoretical motivation ... are not fully verified or supported.**
>
> - Our theoretical motivation stems from the observed distinction in logit patterns between ID and OOD samples: (i) Higher maximum logit values for ID samples; (ii) Higher non-predicted class logits in OOD samples.
>
> - To verify the generality of this phenomenon, we conduct experiments across various ID/OOD datasets and model architectures. Specifically, we report the average maximum logit, and the average logit of non-predicted classes, using ResNet-50, ViT-B/16, and ViT-L/14 as backbone models. As shown in the table below, the results support our hypothesis.
>
> | ResNet50  | max logit | average logit across non-predicted classes |
> | --------- | :-------: | :------------------------: |
> | CIFAR-10 (ID) | 9.020      | -0.1185                    |
> | CIFAR-100 | 7.6920        | 0.3973                     |
> | TIN       | 7.1911       | 0.5673                     |
> | Texture   | 5.5383         | 0.8482                     |
> | SVHN      | 5.5130      | 0.8406  |
>
> | CLIP ViT-B/16 | max logit |  average logit across non-predicted classes |
> | ------------- |  :--------: | :------------------------: |
> | ImageNet (ID) | 30.82     |  17.32                      |
> | iNaruralist   | 26.48     |  17.89                      |
> | SUN           | 26.71     |  17.90                      |
> | Textures      | 28.72     |  18.94                      |
>
> | CLIP ViT-L/14 | max logit | average logit across non-predicted classes |
> | ------------- | :-------: | :----------------------------------------: |
> | ImageNet (ID) |   25.76   |                   11.19                    |
> | iNaturalist   |   20.72   |                   11.65                    |
> | Textures      |   23.85   |                   13.18                    |
> | ImageNet-O    |   21.14   |                   12.16                    |
>
> -----
>
> ### **W1.2: Figure 2 lacks critical context — it is unclear under what setting it was generated.**
>
> Figure 2 was generated using the CLIP ViT-B/16 model, with ImageNet100 as ID dataset and iNaturalist as OOD dataset. The logit curves represent the average logits computed over the entire test set, offering a generalizable summary of the model's behavior rather than illustrating a specific example.
>
> -----
>
> ### **W2 & L3. Limited generalization beyond vision classification tasks.**
>
> While our method is primarily designed for visual classification (a core OOD detection scenario), we also demonstrate its broader applicability. We  evaluate it on  **audio classification**  using  **ESC-50 dataset**, where 50 classes are randomly split into 25 ID and 25 OOD classes. The method is applied with the CLAP [3] model, following the original hyperparameter strategy with $N$ set to 20% of the class count ($N = 5$).
>
> As shown in the table below, our method is also effective beyond vision. Crucially, our method does not rely on modality-specific features or architectures, making it **easily portable to other domains**.
>
> |          | FPR95 $\downarrow$ | AUROC $\uparrow$ | AUPR $\uparrow$ |
> | -------- | ------------------ | ---------------- | --------------- |
> | MCM      | 26.4               | 94.86            | 95.35           |
> | MaxLogit | 40.6               | 92.32            | 92.51           |
> | Energy   | 42.1               | 91.76            | 92.09           |
> | MCM_topN | 27.9               | 94.69            | 95.21           |
> | GEN      | 24.10              | 95.55            | 95.96           |
> | Logitgap | **17.5**           | **96.15**        | **96.48**       |
>
> -----
>
> ### **W3 & L2. Excessive reliance on supplementary material for theoretical explanation.**
>
> We agree that adding clearer intuition in the main text would improve accessibility and we will revise the main paper accordingly. Specifically:
>
> + We will provide a clearer explanation for FPR:  **False Positive Rate (FPR)** measures the likelihood of an ID sample being misclassified as OOD. A lower FPR indicates better OOD detection performance.
> + we will  expand the explanations for Theorem 4.1: Our analysis (Theorem 4.1) shows that LogitGap achieves a lower FPR than MCM when the  temperature $\tau$ exceeds a threshold $z'_1\cdot \frac{K-1}{\ln K}$. This implies that LogitGap better preserves  discriminative logit information, avoiding the over-smoothing effects introduced by softmax in MCM.
>
> Thus, the lower FPR of LogitGap confirms its effectiveness in distinguishing ID from OOD samples compared to MCM.
>
> -----
>
> ### **L1: Motivation Example in Section 4.1 may be unintuitive.**
>
> We agree that the current example might be unintuitive from a classification perspective. In the revision, we will clarify this point by rephrasing Line 139-141 as follows:
>
> *"Since OOD samples tend not to show confident alignment with any known class, sample z2 is more likely to be in-distribution than sample z1. However, the maximum softmax probability for both is identical: ..."*
>
> -----
>
> ### **Q1. The choice subtraction-based formulation in LogitGap needs further justification, with experimental comparisons to alternative designs like weighting the MCM score or other aggregation strategies to strengthen the claim of its effectiveness.**
>
> We subtract the average of remaining logits from the maximum logit because ID samples typically show larger maximum logits and smaller remaining logits than OOD samples. This pattern enhances the ID/OOD distinction, improving detection performance.
>
> We evaluate alternatives on  CLIP ViT-B/16 with ImageNet as ID data:
> 1. MCM_MaxLogit, a weighted combination of MCM and MaxLogit, which improves upon both but is still outperformed by LogitGap (see table below).
> 2. Two LogitGap variants—LogitGap_exp and LogitGap_sqrt—applying exponential and square transformations to amplify score differences. Results show these variants perform comparably to the original LogitGap, indicating different transformations can yield similarly effective discrimination. These comparisons will be included in the revised manuscript.
>
>
> |                     |                    | ninco            |                 |                    | ImageNet-O       |                 |                    | ImageNet-OOD     |                 |                    | Avg              |                 |
> | ------------------- | ------------------ | ---------------- | --------------- | ------------------ | ---------------- | --------------- | ------------------ | ---------------- | --------------- | ------------------ | ---------------- | --------------- |
> |                     | FPR95 $\downarrow$ | AUROC $\uparrow$ | AUPR $\uparrow$ | FPR95 $\downarrow$ | AUROC $\uparrow$ | AUPR $\uparrow$ | FPR95 $\downarrow$ | AUROC $\uparrow$ | AUPR $\uparrow$ | FPR95 $\downarrow$ | AUROC $\uparrow$ | AUPR $\uparrow$ |
> | MCM                 | 79.67              | 73.59            | 95.87           | 75.85              | 79.52            | 98.93           | 80.98              | 78.33            | 85.42           | 78.83              | 77.15            | 93.41           |
> | MaxLogit            | 79.41              | 74.35            | 96.03           | 77.15              | 77.85            | 98.79           | 75.85              | 78.67            | 85.16           | 77.47              | 76.96            | 93.33           |
> | MCM$\times$MaxLogit | 78.31              | 74.76            | 96.12           | 75.70              | 78.96            | 98.87           | 75.77              | 79.36            | 85.95           | 76.59              | 77.69            | 93.65           |
> | LogitGap            | 77.42              | 76.51            | 96.38           | 71.95              | 81.45            | 99.03           | 75.4               | 80.27            | 86.21           | 74.92              | 79.41            | 93.87           |
> | LogitGap_exp        | 77.15              | 76.67            | 96.42           | 71.40              | 81.54            | 99.04           | 74.33              | 80.55            | 86.51           | 74.48              | 79.59            | 93.99           |
> | LogitGap_sqrt       | 77.15              | 76.43            | 96.36           | 71.5               | 81.58            | 99.04           | 75.44              | 80.36            | 86.33           | 74.7               | 79.46            | 93.91           |
>
> -----
>
> ### **Q2. A fair evaluation of MCM should consider N values derived similarly to LogitGap’s.**
>
> To address this concern, we introduce MCM_topN, a comparative baseline that applies the top-N strategy to MCM. For fair comparison, we use the same $N$ for MCM_topN and our LogitGap. Experiments on CIFAR-10 (ID dataset) show that applying the top-N strategy to MCM does not consistently improve performance. Moreover, LogitGap outperforms both MCM and MCM_topN across all 6 benchmarks (see table below).
>
>
> |          | CIFAR100  | TIN      | MNIST     | SVHN     | TEXTURE   | PLACES365 |
> | -------- | --------- | -------- | --------- | -------- | --------- | --------- |
> | MCM      | 87.19     | 88.87    | 92.63     | 91.46    | 89.89     | 88.92     |
> | MCM_topN | 87.22     | 88.65    | 91.87     | 91.12    | 89.61     | 88.63     |
> | LogitGap | **88.01** | **89.7** | **93.44** | **92.1** | **90.73** | **89.74** |
>
>
> [1] A Baseline for Detecting Out-of-Distribution Examples in Image Captioning. ICLR 2017.
>
> [2] Delving into Out-of-Distribution Detection with Vision-Language Representations. NeurIPS 2022.
>
> [3] CLAP: Learning Audio Concepts From Natural Language Supervision. ICASSP 2023.

---

> > ### Comment · Reviewer_4oYE · 2025-08-07
> >
> > **W1.1**
> >
> > The data in Table W1.1 is compelling. I highly recommend adding a new figure to visualize these results, which would strongly reinforce the paper's motivation.
> >
> > This would complement the conceptual illustration in Figure 1 by providing broad empirical validation. For instance, the figure could plot the average scores for MaxLogit, MCM, and LogitGap, comparing the results computed across entire ID versus OOD datasets for various pairings (e.g., ID: CIFAR-10, OOD: CIFAR-100).
> >
> > Such a visualization would clearly demonstrate that LogitGap consistently achieves the largest separation between ID and OOD score distributions across datasets, providing powerful evidence of its robustness.
> >
> > ---
> > **W1.2**
> >
> > As the lack of clear experimental setups pointed out in W1.2 was a common concern among reviewers, please ensure this is thoroughly addressed across the entire manuscript for the final version.
> >
> > ---
> >
> > **W2 & L3**
> >
> > The inclusion of experiments on sound classification is highly commendable. Demonstrating strong performance in this additional domain is a significant result that positively strengthens the paper's contribution.
> >
> > ---
> >
> > **W3 & L2**
> >
> > I suggest that, in addition to the definitions, you also include a concise proof sketch in the main text. This sketch should focus on outlining the key ideas and the logical flow of the proof for your main theorems.
> >
> > ---
> >
> > **L1, Q1**
> >
> > I appreciate your constructive rebuttal. I am confident that the paper will be significantly stronger once the proposed additions are included.
> >
> > ---
> >
> > **Q2**
> >
> > Many readers will likely question whether LogitGap's performance gain comes from its novel formulation or simply from a top-N filtering strategy that could also be applied to MCM. This potential doubt could undermine your core contribution. Your rebuttal experiment comparing LogitGap with MCM_topN compellingly resolves this critical issue. Therefore, I strongly recommend incorporating this specific comparison into the main section where N is introduced to proactively address this reader skepticism.
> >
> > ---
> >
> > **Additional Question**
> >
> > **Q3**
> >
> > After reviewing all the rebuttals, it's clear the new results and clarifications will significantly strengthen the paper. This brings up a practical question: could the authors briefly outline their plan for incorporating this rich new content into the main paper, given the page limits? For example, what existing content might be condensed or moved to the appendix to make space?
> >
> > ---
> >
> > **Q4**
> >
> > While other reviewers have mainly pointed out issues with the captions, I believe the quality and informativeness of the figures themselves could be significantly improved.
> >
> > This becomes especially relevant as you incorporate the new experimental results we've discussed (from Q3). Therefore, could you briefly outline your plan for revising the existing figures or adding new ones?
> >
> > ---
> >
> > **Q5**
> >
> > I presume your extensive experiments with the CLIP encoder are motivated by a deep consideration for the evolving role of OOD detection in the current open-vocabulary era, driven by powerful VLMs.
> >
> > This leads to a fundamental question about the experimental setup. CLIP's encoders already possess knowledge of the classes you've defined as OOD; in fact, an OOD sample can become in-distribution simply by adding its class name to the text prompts.
> >
> > Given this context, do you consider your setup—artificially limiting the class scope for a model that has near-universal knowledge—to be a practical one? If so, could you elaborate on the specific real-world scenarios where this form of OOD detection would be applied?

---

> > > ### Author Response · Authors · 2025-08-08
> > > **Official Comment by Authors 1/2**
> > >
> > > ### **DW1.1. Add a new figure to visualize Table W1.1, and a figure to plot average scores of MaxLogit, MCM, and LogitGap across various ID/OOD pairs.**
> > >
> > > As Figures cannot be submitted during the rebuttal phase, we provide a detailed description of our planned modifications below.
> > >
> > > - Based on Table W1.1, we will plot bar charts of logit statistics for different ID/OOD dataset pairs. For each dataset, two bars will be shown: one showing the **average maximum logit**, and the other showing the **average logit of the non-predicted classes**. We will include a new figure in the *Theoretical Motivation* section to more effectively highlight the motivation behind our method.
> > >
> > > - Also, we will adopt the format of Figure 1 to plot the OOD score distributions of different methods across various ID/OOD dataset pairs. In addition, following Reviewer L4Ge’s suggestion in W1.3, we will include results for the **Energy** method in Figure 1 alongside MaxLogit, MCM, and LogitGap.
> > >
> > > ---
> > > ### **DW1.2. Address the experimental setup issues from W1.2.**
> > >
> > > We will revise the caption of Figure 2 as: "*Descending-sorted logits from CLIP ViT-B/16 on ImageNet100 (ID) and iNaturalist (OOD).*" in the final version.
> > >
> > > ---
> > > ### **DW3 & L2: I suggest that a concise proof sketch should be included in the main text.**
> > >
> > > In the final manuscript, we will add a brief proof sketch for Theorem 4.1 in Section 4.2, as below:
> > >
> > > >**Proof Sketch.**
> > > >
> > > >Theorem 4.1 aims to show that the OOD detection performance of LogitGap is guaranteed to surpass that of MCM.
> > > >To this end, the proof of Theorem 4.1 follows the logical flow outlined below:
> > > >
> > > >1. We begin by introducing an intermediate OOD score function, LogitGap with softmax (LogitGap-softmax). We define it as
> > > >$S_\mathrm{LogitGap-softmax}(\boldsymbol{x};f)=\frac{\sum_{i=1}^K[e^{z_1'/\tau}-e^{z_i'/\tau}]}{K\sum_{j=1}^Ke^{z_j'/\tau}}$, which serves as a bridge between MCM and LogitGap.
> > > >2. We next define the false positive rate (FPR) for different score functions in the following, which measures the performance of score function — a good score function should have a low FPR value. By combining Eq(1) and Eq(2), we have $\lambda_\mathrm{MCM}=\lambda_\mathrm{LM}+\frac{1}{K}$.
> > > >$$FPR_\mathrm{MCM}(\tau, \lambda_\mathrm{MCM})=Q_{x}\left(\frac{e^{z_1'/\tau}}{\sum_{j=1}^ke^{z_j'/\tau}} > \lambda_\mathrm{MCM}\right) (1) $$
> > > >$$FPR_\mathrm{LogitGap-softmax}(\tau, \lambda_\mathrm{LM})=Q_{x}\left(\frac{\sum_{i=1}^K[e^{z_1'/\tau}-e^{z_i'/\tau}]}{K\sum_{j=1}^Ke^{z_j'/\tau}} > \lambda_\mathrm{LM}\right) (2)$$
> > > >$$FPR_\mathrm{LogitGap}(\tau, \lambda_\mathrm{L})=Q_{x}\left(\frac{\sum_{j=1}^K{(z'_1-z'_j)}}{\tau K} > \frac{1}{\tau}\lambda_L\right) (3)$$
> > > >3. Finally, we aim to demonstrate the performance of LogitGap is guaranteed to surpass that of MCM, which is equal to finding the conditions under which the inequality holds: $FPR_{LogitGap}(\tau, \lambda_\mathrm{L}) < FPR_{MCM}(\tau, \lambda_\mathrm{MCM})$. By combining the results from Step 2 and Eq(3), we can derive that the inequality holds with the condition of $\tau > z_1'*\frac{K-1}{\ln K}$.
> > > ---
> > >
> > > ### **DL1, Q1. The proposed additions will significantly strengthen the paper.**
> > >
> > > + In response to **L1**, we will revise Lines 139–141 according to our earlier reply.
> > > + For **Q1**, we will add a subsection in the experiments section to present an ablation study based on our response to Q1. This will compare the average performance of LogitGap and its variants using CLIP ViT-B/16 with ImageNet as the ID dataset, providing further evidence of the method’s effectiveness.
> > >
> > > ---
> > >
> > > ### **DQ2. Incorporate MCM_topN comparison into the main section where N is introduced.**
> > >
> > > We will add a comparison with MCM_topN in Table 5 according to our response to Q2, along with a brief discussion.  We will clarify that LogitGap's advantage goes beyond top-N filtering-it leverages the contrast between the maximum logit and the average non-maximum logits, which better captures ID/OOD separation.
> > >
> > > ---
> > >
> > > ### **DQ4: Could you briefly outline your plan for revising the existing figures or adding new ones?**
> > >
> > > In the revised version, we will:
> > >
> > > - For Figure 1, we will include the result of the Energy method to enable a more comprehensive comparison. Moreover, we will revise the caption to provide detailed information as "_Results are reported using CLIP ViT-B/16 model on ImageNet-20 (ID) and ImageNet-10 (OOD)._"
> > >
> > > - For Figure 2, we will update the caption to clarify the experimental setting (see our response to DW1.2).
> > >
> > > - We will add a new figure (see our response to DW1.1) to visualize the logit statistics across different ID/OOD dataset pairs in the *Theoretical Motivation* section.

---

> > > > ### Author Response · Authors · 2025-08-08
> > > > **Official Comment by Authors 2/2**
> > > >
> > > > ### **DQ3: Could the authors briefly outline their plan for incorporating this rich new content into the main paper, given the page limits?**
> > > >
> > > > Thank you for recognizing the value of our new results and clarifications. Notably, to avoid redundancy and enhance clarity, we address similar concerns raised by multiple reviewers only at their first occurrence. To accommodate these additions within the page limit, we plan to restructure the **main paper** as follows:
> > > >
> > > > + Review 4oYE:
> > > >   + Method
> > > >
> > > >     In Section 4.2, we will: (1) Modify the caption of Figure 2 to include more detailed information; (2) Add a new figure to visualize the logit statistics for different ID/OOD dataset pairs; (3) Add a brief proof sketch to outline the logical flow of the proof; (4) Move Section 4.1 to Appendix B.
> > > >   + Experiments
> > > >
> > > >     **Firstly**, in Section 5.3, we will add a new paragraph and table to compare LogitGap with alternative designs, including MCM_topn, LogitGap_exp and LogitGap_sqrt.
> > > >
> > > >     **Secondly**, we will move the *Hard OOD Detection* subsection of Section 5.2, along with the corresponding Tables 3 and 4, to Appendix C.
> > > >
> > > > + Reviewer L4Ge:
> > > >   + Introduction
> > > >
> > > >     We will revise the Introduction to emphasize that the core assumption of our method is to distinguish ID and OOD samples based on their **logit distribution**.
> > > >   + Experiments
> > > >
> > > >     **Firstly**, we will incorporate the results of GL-MCM and TAG into Table 1 of the paper, based on Table 2.1 from the rebuttal.
> > > >
> > > >     **Secondly**, we will add the results of ViM, GEN, and Energy to Table 5 of the paper, following the findings presented in Table 2.3 of the rebuttal.
> > > >
> > > > + Reviewer AQ8y:
> > > >   + Related Works
> > > >
> > > >     We will add a new paragraph to review related works utilizing logit patterns in both active learning and OOD detection, and summarize several representative methods.
> > > >   + Experiments
> > > >
> > > >     We will provide a more detailed ablation study on the hyperparameter $N$. Specifically, we will add new subfigures to Figure 3 to visualize the FPR values of LogitGap under different $N$ settings across ID/OOD dataset pairs.
> > > >
> > > > In addition, given the many valuable suggestions provided by the reviewers during the rebuttal phase, we plan to include these insights in the **Appendix**.
> > > >
> > > > + Review 4oYE:
> > > >
> > > >   We will add a new subsection in Appendix C to discuss the broader applicability of LogitGap. Specifically, we will add a new table to represent experimental results of LogitGap on an audio classification task.
> > > >
> > > > + Review BQ18:
> > > >
> > > >   We will add another subsection in Appendix C to empirically demonstrate the effectiveness of using synthetic OOD data for selecting suitable hyperparameter $N$.
> > > >
> > > > + Reviewer L4Ge:
> > > >
> > > >   **Firstly**, for the traditional OOD detection setting, we will add experimental results on ImageNet using ResNet-50, Swin-T, and ViT-B/16.
> > > >
> > > >   **Secondly**, for the zero-shot OOD detection setting, we will include results on ImageNet using CLIP ViT-L/14 and CLIP ResNet-101.
> > > >
> > > > + Reviewer AQ8y:
> > > >
> > > >   **Firstly**, we will add a new section in Appendix to discuss the differences between LogitGap and some closely related methods (such as margin sampling), along with an empirical comparison.
> > > >
> > > >   **Secondly**, we will add a new subsection in Appendix C to discuss the limitations and potential negative impacts of our LogitGap in the few-shot OOD setting, especially on ImageNet-R.
> > > >
> > > > ---
> > > > ### **DQ5: Do you consider zero/few-shot OOD setup to be a practical one? If so, could you elaborate on the specific real-world scenarios where this form of OOD detection would be applied?**
> > > >
> > > > We fully agree that, with the increasing popularity of VLMs like CLIP in the open-vocabulary era, the traditional OOD detection setup requires reconsideration in a more complex context.
> > > >
> > > > - **The significance of zero-shot OOD detection paradigm**: Traditional OOD methods typically rely on task-specific models that require separate training for each dataset or task, which limits their adaptability in real-world scenarios where tasks evolve frequently.  In contrast, our method depends only on a set of ID class names and can perform OOD detection across different tasks based on the CLIP model, demonstrating stronger generality and practical value.
> > > >
> > > > - **Specific scenarios: Task-defined restricted class sets.**  Although CLIP models have broad general knowledge across many categories, in practical deployments, practical deployments often necessitate restricting the model's inference scope.  In applications such as **medical diagnosis**, **industrial inspection**, and **security monitoring**, systems usually recognize only a well-defined set of classes to ensure prediction **reliability and safety**. Even though CLIP has seen a broader set of classes during pretraining, samples belonging to classes outside the current task's scope should still be regarded as OOD.
> > > >
> > > > In summary, zero/few-shot OOD detection with CLIP is practical and better meets real-world needs like task flexibility, limited training, and deployment efficiency.

---

> > > > > ### Comment · Reviewer_4oYE · 2025-08-08
> > > > >
> > > > > The kind and detailed answers and the concrete plan were very impressive. All of my questions have been addressed.
> > > > >
> > > > > I look forward to an even better paper in the final version, with specific details on points like the "additional figure & caption," the "brief logical flow for Theorem 4.1," and "the significance of OOD in the open-vocabulary era -> why the experiments were primarily conducted with the CLIP encoder," along with a clear organization of the additional experiments you conducted.
> > > > >
> > > > > Many of the weaknesses and questions were resolved during the rebuttal process, and I have come to agree more with the excellence of this paper. Therefore, I will be raising my score.

---

> > > > > > ### Author Response · Authors · 2025-08-08
> > > > > > **Official Comment by  Authors**
> > > > > >
> > > > > > Thank you again for your kind response and for taking the time to review our rebuttal. We truly appreciate your constructive feedback in helping to improve our work.

---

### Official Review · Reviewer_BQ18 · 2025-07-14

**Clarity:** 4
**Significance:** 3
**Originality:** 3
**Rating:** 4
**Confidence:** 4

**Summary:**

LogitGap is a simple post-training method for differentiating between in-distribution (ID) and out-of-distribution (OOD) samples during inference. LogitGap uses a scoring function that explicitly measures the average gap between a sample’s largest logit and its remaining logits, exploiting the observation that ID samples have a much larger 'logit gap' than OOD samples. Using the logits from all K classes in the scoring function can increase noise in the score as many classes that are semantically far from the sample have negligible logit values. Hence, the authors compute the gap only over the top-N logits. N is chosen automatically from a handful of ID samples to maximize ID/OOD separation. A theoretical analysis links LogitGap to softmax–based scores such as MCM and proves it yields lower false-positive rates when the softmax temperature is high, while extensive experiments confirm the practical gains. On zero-shot CLIP ViT-B/16, LogitGap cuts FPR95 by up to 5.8% versus the previous best logit method, and when paired with few-shot prompt-tuning or training-time OOD schemes it delivers consistent extra improvements across semantic-shift and covariate-shift benchmarks.

**Questions:**

- Can LogitGap be applied to features at an earlier layer in the DNN in order to detect an OOD sample wtihout performing the full inference computation? This would further reduce the overhead of OOD inference.

**Ethical Concerns:**

["NO or VERY MINOR ethics concerns only"]

**Limitations:**

yes

**Quality:**

3

**Strengths And Weaknesses:**

Strengths

- The scoring function is simple to compute given the logits and does not require additional training.
- LogitGap scoring function uses information from all logits as opposed to prior methods that may use only the maximum value.
- The false-positive rate has a minimum when N lies between 20% and 50% of classes, thereby simplifying selection of N for image classification tasks.

Weaknesses

- We must perform the computation for all DNN layers in order to obtain the logits to determine whether a sample is ID or OOD.
- Determining the correct N is not entirely free as LogitGap requires access to a small ID validation set, which may be unrealistic in zero-sample deployment scenarios
- Relies on synthetic OOD samples generated by noise/interpolation, whose representativeness for real-world OOD samples is unclear and not justified.

---

> ### Author Rebuttal · Authors · 2025-07-31
>
> We sincerely thank the reviewer for the insightful comments. Below, we address each concern.
>
>   ### **W1. The LogitGap requires the computation for all DNN layers to obtain the logits to determine whether a sample is ID or OOD.**
>
>   We would like to clarify that OOD detection is typically performed alongside the primary classification task, whose goal is to identify OOD samples without degrading ID classification accuracy [1]. In this context, the logits required for OOD detection are **already produced** during the standard forward pass of classification. Therefore, our method leverages these existing computations and **adds no extra computational cost** beyond standard inference.
>
>   [1] Generalized Out-of-Distribution Detection: A Survey, IJCV 2024.
>
> ------
>
>   ### **W2. Determining the correct N is not entirely free, as LogitGap requires access to a small ID validation set, which may be unrealistic in zero-sample deployment scenarios.**
>
>   We would like to clarify that our proposed method can be applied in the zero-sample scenario.
>   Specifically, our method includes two variants (Lines 230-232): **LogitGap** (fixed N = 20% of classes) and **LogitGap\*** (adaptive N via ID validation set). As shown in Tables 1 and 2, LogitGap performs comparably to LogitGap\*, demonstrating that our method remains effective even without access to a validation set.
>
> ------
>
>   ### **W3. Relies on synthetic OOD samples generated by noise/interpolation, whose representativeness for real-world OOD samples is unclear and not justified.**
>
>   Thanks for this valuable comment. We agree that synthetic OOD samples may not fully reflect the characteristics of real-world OOD samples. However, our empirical results show that these synthetic samples are sufficiently informative  for selecting a robust hyperparameter $N$.
>
>   As shown in the table below, on ImageNet, the optimal $N$ selected using synthetic samples (i.e., $N_{syn}$) is close to the optimal $N$ derived from multiple real OOD datasets (i.e., $N_{real}$), and achieves comparable performance. This demonstrates the practicality and generalizability of our strategy for selecting $N$ without requiring access to real OOD data.
>
>   |          |       | ninco |      |       | Imagenet-O |      |       | ImageNet | -OOD     |
>   | -------- | ----- | ----- | ---- | ----- | ---------- | ---- | ----- | ------------| ---- |
>   |          | FPR95 | AUROC | N    | FPR95 | AUROC      | N    | FPR95 | AUROC        | N    |
>   | $N_{syn}$ | **77.42** | **76.51** | 88   | **71.95** | 81.45      | 88   | **75.40** | **80.27**        | 88   |
>   | $N_{real}$     | 77.22 | **76.51** | 100  | 71.60 | **81.50**      | 110  | 75.39 | **80.27**        | 90   |
>
> -----
>
>   ### **Q1: Can LogitGap be applied to features at an earlier layer in the DNN in order to detect an OOD sample without performing the full inference computation?**
>
>   Detecting OOD samples using only shallow features is challenging. In zero/few-shot OOD detection settings, OOD samples often involves **semantic shifts**, which are difficult to capture without the high-level abstractions provided by deeper layers [2].
>
>   To address this, existing works [3, 4] explicitly leverage deep features to capture high-level semantic cues for OOD detection.
>   However, these methods may incur additional computational overhead as they require calculating feature correlations for each layer.
>   In contrast, our approach remains computationally efficient while still effectively identifying OOD samples.
>
>   [2] Full-Spectrum Out-of-Distribution Detection. IJCV 2023.
>
>   [3] A simple unified framework for detecting out-of-distribution samples and adversarial attacks. NeurIPS 2018.
>
>   [4] Detecting out-of-distribution examples with gram matrices. ICML 2020.

---

> > ### Comment · Reviewer_BQ18 · 2025-08-09
> >
> > Thank you for your detailed rebuttal. All of my questions have been addressed

---

### Comment · Area_Chair_rymq · 2025-08-04
**Author-Reviewer Discussion**

Dear Fellow Reviewers,

The authors have provided detailed replies to your comments. They also managed to provide some additional empirical results, e.g., on ImageNet-O/OOD, using average maximum logit vs. the average logit, audio classification, more zero-shot/traditional OOD detectors as baselines, etc.

Please take a look at the authors' rebuttal and all the reviews, and start to engage with the authors as early as possible to discuss your concerns. Your active participation in this stage would mean a lot to the authors.

Thank you!

AC, NeurIPS 2025

---

### Note · Authors · 2025-08-13

We sincerely thank all reviewers for their time and constructive feedback during the rebuttal phase.
We are encouraged that all questions from **Reviewers BQ18, 4oYE, and AQ8y** have been addressed, including computational efficiency, hyperparameter selection, related works discussion, writing refinements and theoretical explanation.

Specifically, for **Reviewer L4Ge**, we have addressed most issues raised in the initial review (W1.1-W2.3, W2.6 and W2.7), including experiments with additional baselines, datasets, and different model architectures.
For the other remaining concerns (Second Round D1 and D2 in Discussion-Phase), we would like to clarify further in the final remarks.

- Second Round D1 requests the implementation details for a baseline method.
  While we have clarified this in our response to W2.3, it seems the details may have been overlooked, leading to a misunderstanding.
  To address this, we reply with a more detailed explanation along with additional experiments.
- Second Round D2 suggests some manuscript revisions about our contribution.
  In response, we emphasize the corresponding lines in the main paper and explain how the manuscript would be updated.

**Although we did not receive further feedback or see adjustments to the score from Reviewer L4Ge, we believe our responses address these concerns adequately.**

---

### Decision · Program_Chairs · 2025-09-17

**Decision:**

Accept (poster)

**Comment:**

The work proposes a new logit-based scoring method called LogitGap for OOD detection in vision-language models. The method shows effective performance under several challenging scenarios, such as zero-/few-shot setting and hard OOD detection, on some popular ID/OOD benchmarks.

Strengths of the work include: 1) simple yet effective method LogitGap, 2) good theoretical analysis and extensive empirical justification, and 3) well-organized paper.

There were a number of major concerns on technical novelty, reliance on validation set with ID (and synthetic OOD data) samples, unverified theoretical assumption, generalization beyond vision tasks, missing baselines, experiments limited to small ID/OOD benchmarks (e.g., missing results on ImageNet-1k data), outdated zero-shot detection contenders, insufficient ablation study.

After author rebuttal and author-reviewer discussion, all four reviewers confirmed that their main concerns have been adequately addressed, three of which increased their ratings to borderline accept. This leads to four border accept recommendations.

One key remaining concern is that the revisions required to be made are rather significant. The authors have promised on these changes, and should make every possible effort to accommodate all the requested revisions if the paper is accepted.